# Kallikrein-8 mediates furin-independent Activin-A precursor processing to stimulate tumor growth in melanoma

Manon Bulliard [1], Katarina Pinjusic [1,3], Laura Iacobucci[1], Céline Schmuziger [1], Nadine Fournier [1,2] & Daniel B. Constam [1] ✉

Receptor binding of TGF-β and related ligands such as Activin-A requires cleavage of a furin site in their dimeric precursor proteins. Melanoma cells cleave one Activin-A subunit independently of furin and related proprotein convertases, raising questions of how this half-processed intermediate is generated and whether it influences tumor growth. Here, an siRNA library screen for proteases mediating this furin-independent "hemicleavage" identifies kallikrein (Klk)-8. While a KLK8 cleavage site in proActivin-A overlaps with the furin recognition sequence, its exposure is limited and requires prior transient acidification. Therefore, only furin efficiently converts proActivin-A to fully mature form both in tumor cells and in cell-free cleavage assays. Moreover, knockdown of *Klk8* in syngeneic melanoma grafts suppresses Activin-A induced tumor growth, demonstrating that cleavage by only furin is not sufficient. Besides elucidating how Activin-A processing is regulated, our findings show that KLK8 holds promise as a target to mitigate Activin-A induced tumor growth.

Activin A (hereafter Activin-A) is a multifunctional secreted protein of the transforming growth factor (TGF)-β family that can activate SMAD2 and SMAD3 transcription factors and SMAD-independent signal transduction pathways by assembling heterooligomeric complexes of activin receptor (ActR)-IB with ActR-IIA or -IIB[1]. Activin receptors are widely expressed in multiple cell types and can inhibit or promote cancer progression by regulating cell proliferation, migration, or differentiation in healthy tissues and in many cancer types[2]. Depending on the context, Activin-A can also locally enhance or diminish tissue inflammation, or accumulate in the circulation where it promotes a cancer-associated systemic muscle wasting syndrome known as cachexia[3,4]. In the skin, Activin-A is implicated in fibrosis[5] and in promoting squamous cell carcinoma[6,7]. Furthermore, treatment of cultured skin melanocytes with exogenous Activin-A in amounts exceeding the secreted antagonist follistatin (FST) led to cell cycle arrest, raising the possibility that activin signaling in these cells may

have a tumor-suppressive function[8]. In line with this prediction, lentiviral transduction of B16-F1 mouse melanoma cells with a ligand-independent mutant form of the activin receptor ActR-IB diminished tumor growth in syngeneic recipient mice[9]. However, a transgene encoding secreted Activin-A in the same melanoma model not only failed to slow primary or metastatic tumor growth, but even stimulated it by blunting T cell-mediated anti-tumor immunity[9]. This suggests that autocrine signaling of secreted Activin-A was too weak or insufficient to prevent a net increase in tumor growth induced by cell non-autonomous effects on the tumor microenvironment. Subsequent validation in this and other melanoma models confirmed that Activin-A secretion by melanoma cells impairs adaptive anti-tumor immunity and the response to adoptive T cell transfer and immune checkpoint blockade therapies, correlating with an indirect inhibitory effect on tumor infiltration by cytotoxic CD8[+] T cells[10]. Activin-A is frequently expressed in human melanoma and can confer a selective advantage to

[1]École Polytechnique Fédérale de Lausanne (EPFL) SV ISREC, Station 19, 1015 Lausanne, Switzerland. [2]Translational Data Science (TDS) facility, Agora Cancer Research Center, Swiss Institute of Bioinformatics (SIB), Bugnon 25A, 1015 Lausanne, Switzerland. [3]Present address: Dana-Farber Cancer Institute, 450 Brookline Avenue, Boston, MA 02215, USA. ✉e-mail: Daniel.Constam@epfl.ch

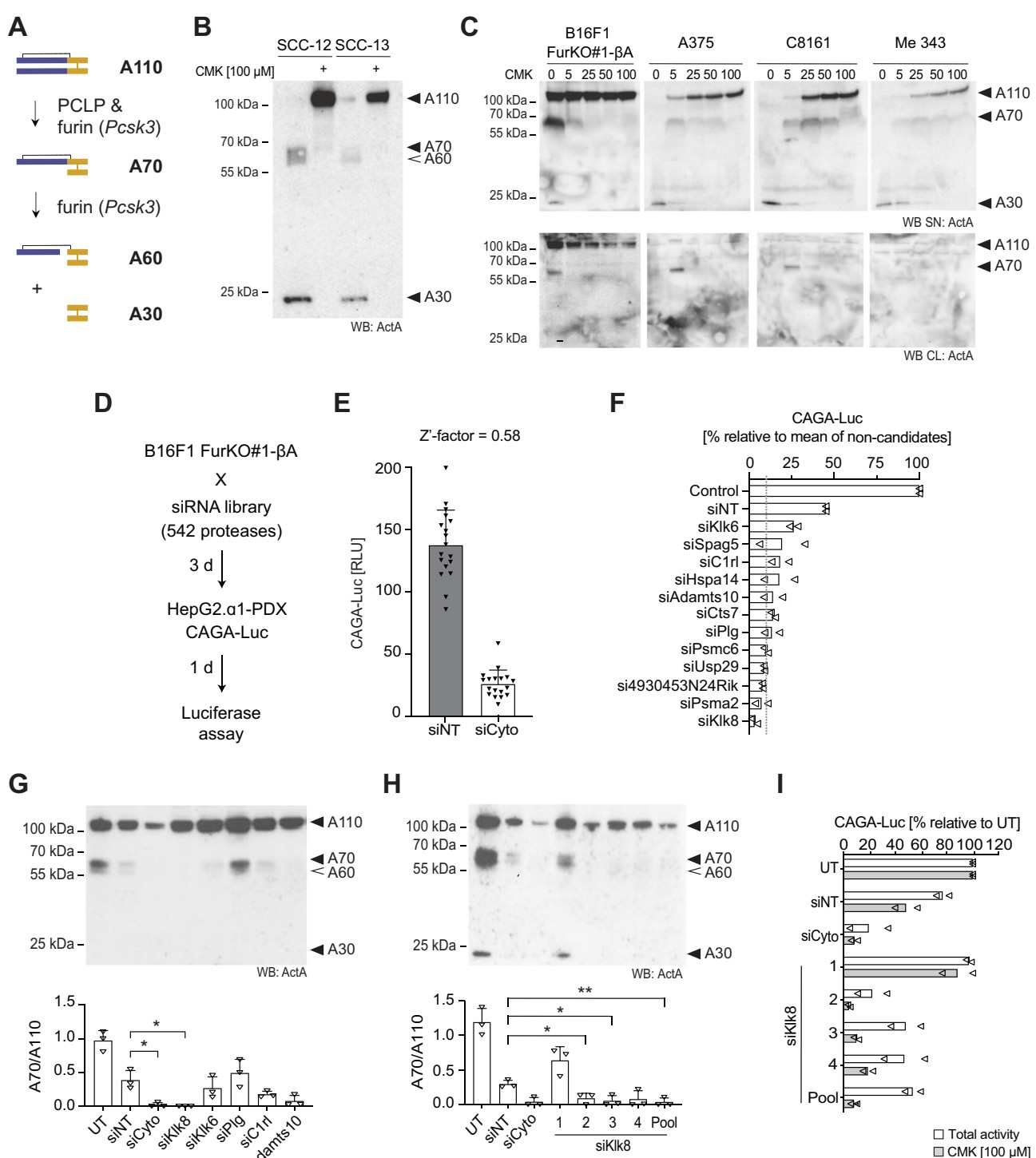

**Fig. 1 | RNAi screening in B16-F1 melanoma cells identifies Klk8 as a protease that mediates furin-independent Activin-A maturation. A** Schema of Activin-A processing by furin and PC-like protease (PCLP). Violet, prodomain; yellow, disulfide-linked dimer of the mature region; stippled line, disulfide bond linking mature region to one prodomain. **B** Activin-A Western blots of cell supernatants (SN) of SCC-12 and SCC-13 cells with or without 100 μM CMK for two days. Data represent three experiments. **C** Activin-A in SN and lysates (CL) of A375, C8161, Me 343, and *INHBA*-transduced *Furin* knockout (FurKO#1-βA) B16-F1 cells after two days of conditioning with different CMK doses. Data are representative of three experiments. **D** Strategy of the siRNA library screen for PCLP activities that mediate hemicleavage of βA-encoded proActivin-A in B16F1 FurKO#1-βA cells. **E** Z'-factor representing the effect size of cytotoxic (siCyto) versus non-targeted (siNT) siRNA cocktails in FurKO#1-βA cells on CAGA-Luc expression in co-cultured HepG2.α1-PDX reporter cells (n = 18 technical replicates). Values are means ± SD (Two-sided Student's t-test). **F** Inhibition of Activin-

A signaling by siRNA cocktails targeting the top-scoring PCLP candidates. Data are normalized to total Activin-A and represent the means from two biological replicates per siRNA. Stippled gray line, arbitrary threshold of 5-fold reduction relative to siNT. **G** Activin-A Western blot of B16F1 FurKO#1-βA cell SNs 3 d after transfection with 10 nM of the indicated siRNA pools. Average A70/A110 ratios from three experiments are quantified below the blot. UT, untransfected. **H** As in (**G**), but after transfection with individual *Klk8* siRNAs (10 nM). **I** CAGA-Luc induction in HepG2 reporter cells treated for 12 hrs with SNs of siRNA-transfected B16F1 FurKO#1-βA cells. To determine the percentage of signal inhibition by the siRNAs, values were normalized to Activin-A signaling activity in SN of untransfected (UT) cells. To distinguish how much of this remaining "total activity" was due to Activin-A cleavage in the melanoma cells, further proteolytic processing during the incubation was blocked by treating reporter cells with the PC inhibitor dec-RVKR-cmk (CMK, 100 μM) where indicated. Data represent the means of two experiments (Two-sided Student's t-test). *p < 0.05, **p < 0.01.

cancer cells when they find a way to evade its tumor-suppressive activity[9]. However, therapeutic strategies to safely target Activin-A, or to block a shift from tumor-suppressive towards tumor-promoting signaling activity are unavailable.

Activin-A derives from a precursor of two inhibin beta A (βA) subunits encoded by the *INHBA* gene that homodimerize in the endoplasmic reticulum to form pre-proActivin-A[1]. Following the removal of the secretory signal sequence, proActivin-A must be cleaved after a conserved penta-arginine motif to enable a disulfide-linked homodimer of only the C-terminal mature region comprising residues 311-426 to bind activin receptors[11,12]. The stretch of five arginines at the cleavage site of Activin-A fits the consensus recognition motif of several basic residue-specific proprotein convertases (PCs) of the subtilisin/kexin (PCSK) family, including furin, PACE4, PC5/6, and PC7. These PCs are broadly expressed in many cells and tissues and capable of cleaving a plethora of proproteins in constitutively secreted vesicles of the *trans*-Golgi network (TGN), in the extracellular space, or in endosomes[13]. Typically, they cleave their substrates after the recognition sequence R-X-K/R-R, where X can be any residue other than cysteine, in contrast to PC1 and PC2/3 which specifically recognize dibasic KR or RR motifs of substrates in the regulatory exocytic apparatus of neuroendocrine cells, or PCSK8 or PCSK9 which do not cleave after basic amino acids[14]. Previous analysis in *Furin* knockout (FurKO) B16-F1 mouse melanoma cells established that loss of furin in these cells suffices to largely suppress the production of mature Activin-A and to stabilize the 110 kDa precursor (A110) in culture medium. Interestingly, a processing intermediate migrating with an apparent molecular weight of 66 kDa was stabilized alongside and formed even independently of furin[15]. An analogous hemicleaved Activin-A consisting of a dimer of one cleaved and one uncleaved βA subunit was initially described as a protein of almost 70 kDa protein also in other cell types[11,12]. Therefore, we referred to hemicleaved Activin-A as A70, whereas a more mobile 58 kDa form and the fully mature Activin-A (26 kDa) secreted by *Furin* wild-type cells were named A60 and A30, respectively[15,16]. Analysis on reducing gels and by mass spectrometry confirmed that one βA subunit of A70 was uncleaved, whereas A60 consists of a mature Activin-A isoform that is covalently linked to the prodomain of one of its cleaved subunits via an intramolecular disulfide bond between cysteines C35 and C314 (Fig. 1A)[16]. Importantly, A70 formation was also maintained in double knockout (DKO) cells lacking both furin and PC7, the only other known basic residue-specific PCSK activity expressed in B16-F1 melanoma cells or in human skin melanocytes[15,17]. Therefore, we wondered if furin- and PC7-independent hemicleavage of proActivin-A relies on a PC-like protease (PCLP) distinct from known PCSK family members.

Here, we conduct an siRNA library screen to identify the PCLP activity mediating proActivin-A hemicleavage in FurKO and DKO cells, and to test its function in a syngeneic melanoma grafting model and by analyzing human cancer databases. Among several candidates, we show that depletion of kallikrein−8 (Klk8) inhibits furin-independent hemicleavage in vitro as well as the Activin-A-induced tumor growth advantage in vivo, and that a point mutation in the furin motif of proActivin-A that blocks hemicleavage by recombinant KLK8 has a similar protective effect. In addition, we show that *KLK8* expression in this and several other human cancers correlates with poor survival, further adding to its appeal as a future therapeutic target.

## Results

### RNAi screening for candidate PCLP activities identifies KLK8
Previous analysis in furin knockout (FurKO) or furin and PC7 double knockout (DKO) CRISPR clones of B16-F1 mouse melanoma cells transfected with an *INHBA* (βA) expression vector established that the conversion of Activin-A precursor (A110) to fully mature 26 kDa form (A30) relies on furin (Fig. 1A). To assess by a CRISPR-independent approach if only furin is essential, siRNA pools targeting this or other

PCs were transfected into B16-F1 cells that stably express *INHBA* as a lentiviral transgene (B16F1-βA). Of the *Pcsk* transcripts detected by RT-qPCR analysis, all were depleted by more than 90% by the corresponding siRNA pools relative to non-targeted control siRNA (siNT), including *Pcsk3*/furin, *Pcsk7*/PC7, and *Pcsk9* (Fig. S1A). Analysis of the cell supernatants (SN) by immunoblotting on non-reducing gels revealed that *Pcsk3* knockdown but no other Pcsk siRNAs stabilized the 110 kDa Activin-A precursor (A110) (Fig. S1B). A70 was stabilized alongside, confirming that it forms independently of furin, as described previously in FurKO and DKO cells (Fig. 1A)[15]. Of note, mature Activin-A (A30) was barely detected even in control cells receiving only siNT, because under RNAi conditions, furin instead converted Activin-A mainly to the alternative 58 kDa mature form A60, possibly reflecting increased ROS formation in cells treated with cationic liposomes[18]. The corresponding alternative isoforms were also formed by endogenous Activin-A in the human squamous cell carcinoma cell lines SCC-12 and SCC-13, and in human A375, Me 343, and C8161 melanoma cells (Fig. 1B, C). Treatment with increasing concentrations of the membrane-permeable pan-PC inhibitor decanoyl-RVKR-chloromethylketone (CMK) inhibited A70 and A60 formation, confirming that they arise by proteolytic processing.

To assess the effect of Activin-A processing on SMAD3 signaling, we incubated the SNs of the siRNA-transfected B16F1-βA cells on HepG2 reporter cells stably expressing the SMAD3 luciferase reporter CAGA-Luc and, for signal normalization, CMV-Renilla (Fig. S1C)[19]. Under these conditions, an increase in the ratio of CAGA-Luc expression relative to CMV-Renilla measures the total activity of Activin-A released by the sum of precursor cleavage by the melanoma cells themselves plus any further proteolytic processing by the signal-receiving HepG2 reporter cells. To prevent further Activin-A processing in the conditioned medium during its incubation on HepG2 reporter cells, the latter were co-treated with CMK[20]. Alternatively, or in addition, they were stably transduced with the furin-inhibitory antitrypsin variant α1-PDX[21]. While CMK treatment of the reporter cells diminished the induction of CAGA-Luc by approximately 50%, transfection of the signal-sending melanoma cells with *Pcsk3* siRNA but not RNAi of other PCs increased this inhibition to 75% (Fig. S1C). To validate that reporter cells contribute specifically to Activin-A precursor processing, HepG2 reporter cells with or without α1-PDX were co-cultured with βA-transduced FurKO cells in the presence or absence of CMK (Fig. S1D). Western blot analysis of Activin-A in SN of these co-cultures revealed that α1-PDX expression in reporter cells dramatically stabilized A110 at the expense of A60 and A30 formation, confirming that signal-receiving reporter cells significantly cleave melanoma cell-derived A110 and A70. However, to block A70 formation within melanoma cells and to completely inhibit signaling in the HepG2.α1-PDX reporter cells, these co-cultures in addition required treatment with CMK. These data show that A70 formation by the PCLP activity in melanoma cells can be blocked by adding CMK, but not by α1-PDX from the signal-receiving HepG2.α1-PDX cells.

To screen for proteases mediating A70 formation in melanoma cells independently of *Furin*, a B16-F1 FurKO clone that stably expresses *INHBA* (FurKO#1-βA) was transfected with siRNA pools against 542 known proteases, followed by co-culture with HepG2.α1-PDX CAGA-Luc reporter cells (Fig. 1D). Negative and positive controls to optimize the screening conditions consisted of cocktails of cytotoxic siRNAs (siCyto) targeting essential cell survival genes, and of a commercial non-targeted siRNA (siNT), respectively. Signal normalization to total Activin-A secreted by viable cells (i.e. including precursor) served to distinguish real candidates from false positives. At cell densities optimized for 96-well format, transfection of B16F1 FurKO#1-βA cells with siCyto reduced the induction of CAGA-Luc in co-cultured reporter cells 5.5-fold compared to siNT control, and with a Z' factor of 0.58 (Fig. 1E). Only twelve of the 542 siRNA pools examined reduced the normalized signal below an arbitrary threshold of 5-fold relative to the mean of all

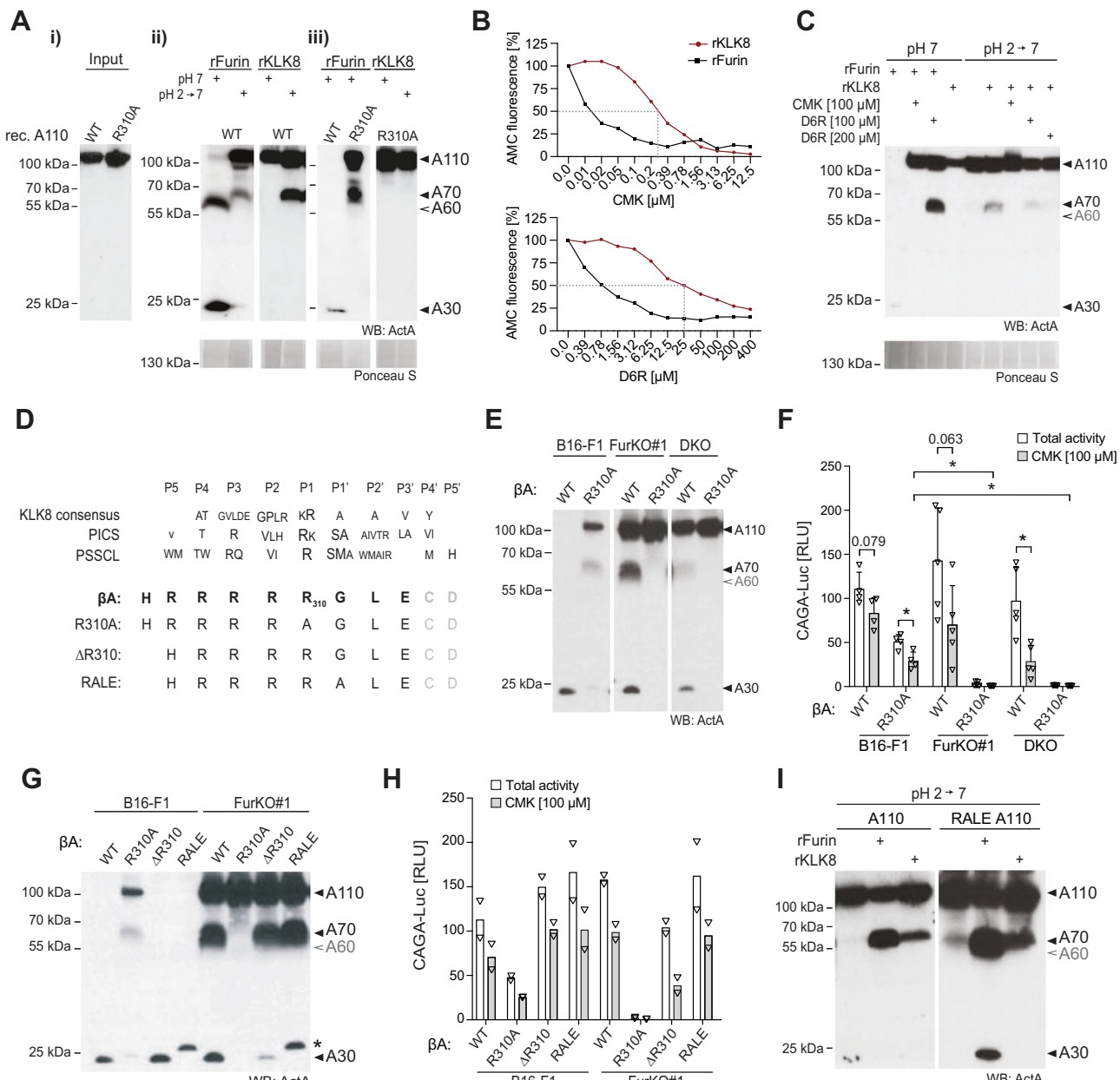

**Fig. 2 | Alanine substitution of arginine 310 or treatment with furin inhibitors abrogates Activin-A hemicleavage by kallikrein 8. A** Wild-type (WT) or R310A mutant proActivin-A (A110) analyzed (i) before or (ii and iii) after incubation with recombinant (rFurin) or rKLK8 for 5 hrs at pH 7. Where indicated, these SNs were first transiently acidified to pH 2 for 60 min. Results are representative of three experiments. **B** Effect of increasing concentrations of dec-RVKR-cmk (CMK) or D6R on the in vitro cleavage of Boc-VPR-AMC or Boc-RRVR-AMC by rKLK8 or rFurin, respectively. Data represent the mean of two experiments. **C** WT A110 treated as in (**A**), but with or without CMK or D6R. Data represent two experiments. **D** Sequence comparisons of the Activin-A precursor cleavage site with the KLK8 consensus motif in the MEROPS database. The KLK8 substrate specificity determined by proteomic identification of cleavage sites (PICS) or by a positional-scanning screening of a synthetic diverse tetrapeptide library (PSSCL) is shown below[22]. **E** Activin-A Western blot of SNs from B16-F1, FurKO#1 and DKO cells

cultured for 48 hrs after transfection with WT or R310A mutant βA. Data represent three experiments. **F** Induction of CAGA-Luc in HepG2 reporter cells within 12 hrs after adding the SNs from the samples analyzed in (**E**). Where indicated, the HepG2 reporter cells were treated with 100 μM CMK. Data represent the means ± SD of four to five experiments (One-way ANOVA). **G** Activin-A Western blot of SNs from B16-F1 and FurKO#1 cells cultured for 48 hrs after transfection with WT-βA, R310A-βA, ΔR310, or RALE-βA. Asterisk, size-shifted RALE-mutant A30. Data represent three experiments. **H** Induction of CAGA-Luc in HepG2 reporter cells within 12 hrs after adding the SNs analyzed in (**G**). Where indicated, the HepG2 reporter cells were treated with 100 μM CMK. Data represent the means of two experiments. **I** Activin-A Western blot of wild-type or RALE mutant pro-Activin-A (A110) containing SNs that were transiently acidified to pH 2 for 60 min, followed by incubation for 5 hrs with rFurin or rKLK8. Results are representative of two experiments. *$p < 0.05$.

non-candidate siRNA pools (Fig. 1F). Transfection of siNT also reduced signaling, albeit only approximately 2-fold, correlating with a non-specific reduction of total Activin-A, and with the position of this siRNA at the border of the plate. CAGA-Luc induction also decreased more than 5-fold if the FurKO-βA cells were transfected with *Pcsk2* siRNA

(Fig. S1E). However, this decrease and a similar trend for *Pcsk1* siRNA may be non-specific because *Pcsk1* and *Pcsk2* mRNAs were below detection in these cells, and their targeting by RNAi did not inhibit A70 formation (Fig. S1F). Therefore, we prioritized other candidates that we selected based on literature about their substrate specificity, tissue

distribution, or known links to diseases. To validate PCLP activity, we tested by Western blot analysis whether RNAi of the selected candidates specifically decreases the ratio of A70 relative to uncleaved precursor (A110) in melanoma cell SNs. Among the top candidates, especially *Klk8* significantly decreased the A70/A110 ratio in FurKO#1-βA cell SNs (Fig. 1G). Three siRNAs in the *Klk8* siRNA pool similarly decreased the A70/A110 ratio also when transfected individually (Figs. 1H and S1G). Importantly, correlating with this depletion of hemicleaved Activin-A, the *Klk8* siRNA pool and individual siRNAs also inhibited CAGA-Luc induction by the same SNs in CMK-treated HepG2 reporter cells (Fig. 1H, I). By contrast, total activity in reporter cells without CMK was not significantly depleted, even though *Klk8* RNAi tended to also diminish the levels of proActivin-A and its signaling activity, correlating with a decrease in viable melanoma cells (Fig. S1H). These data indicate that *Klk8* RNAi inhibits furin-independent Activin-A hemicleavage and signaling. To directly validate knockdown efficiency, we transfected *Klk8* siRNAs into YUMM3.3 mouse melanoma cells since *Klk8* mRNA levels in B16-F1 cells were too low for reliable quantification (Fig. S1I, J). Analysis by RT-qPCR after 72 hrs showed that all siRNA sequences and their pool reduced *Klk8* mRNA levels in this cell line by approximately 50% (Fig. S1K). However, siKlk8_1 targets the 3'end of the coding sequence that is missing in an alternative transcript, explaining perhaps why it failed to inhibit the function of Klk8 in B16-F1 cells.

## KLK8 mimics PCLP activity in a cell-free assay and is sensitive to inhibitors of basic PCs

To further verify cleavage by KLK8, we expressed proActivin-A in a distinct FurKO CRISPR clone (clone #3) where processing by endogenous PCLP activity fortuitously was inhibited (Fig. 2A, panel (i)). Control treatments of the SN from the resulting FurKO#3-βA cell line with recombinant furin (rFurin) at pH 7 converted A110 to A60 and A30, as expected. By contrast, incubation with rKLK8 only generated A70, and only if A110 was first exposed to transient acidification to pH 6.5 or below (Fig. 2A panel (ii), and Fig. S2A, B). Interestingly, transient acidification also enabled hemicleavage by rKLK5, but not by the more distantly related KLK6 or KLK12 (Fig. S2A), even though all four kallikreins activated their control substrates (Fig. S2C). However, *Klk5* expression in B16F1 FurKO#1-βA cells was below detection, and not required for signaling (Fig. S2D, E). This likely explains why Klk8 was the only kallikrein mediating PCLP activity in our RNAi screening assay. The rationale for transient acidification of the SNs was to address if the conformation of proActivin-A or interacting factors influence the access of furin or KLK8. Whereas the cleavage by rKLK8 and rKLK5 increased after acid treatment, processing by rFurin decreased, leading to inefficient A110 cleavage and failure to convert A70 to A60 or A30 (Fig. 2A (ii)). In the absence of acid treatment, we noticed that the treatment with rFurin diminished the amount of Activin-A in some batches of cell-conditioned media, pointing to likely degradation or decreased solubility (Fig. 2A (iii)). Regardless of the mechanism, these results show that transient acidification favors hemicleavage by KLK8 but diminishes Activin-A maturation by furin.

Hemicleavage of proActivin-A can be suppressed by deleting its conserved furin recognition sequence RRRRR$_{310}$[11]. If KLK8 directly binds this multibasic motif, furin inhibitors mimicking this sequence should also inhibit KLK8. In line with this prediction, incubation with CMK containing the peptide sequence RVKR, or with hexa-D-arginine (D6R), inhibited the cleavage of fluorogenic control substrates by rFurin as well as rKLK8 (Fig. 2B). The half-maximal inhibitory concentration (IC50) of D6R for rKLK8 was below 25 μM, compared to 0.78 μM for rFurin. rKLK8 was also less sensitive than rFurin to inhibition by CMK, the IC50 values being 0.28 or 0.015 μM, respectively (Fig. 2B). Importantly, both inhibitors also diminished rKLK8-mediated Activin-A hemicleavage (Fig. 2C). These results are consistent with the notion that KLK8 may directly cleave proActivin-A at the multibasic furin recognition motif.

## Activation of proActivin-A by KLK8 depends on arginine 310

To determine at which residues KLK8 can cleave the furin motif of proActivin-A, we incubated a 41 amino acid peptide comprising this motif and its flanking sequences with recombinant proteases. Analysis by mass spectrometry showed that treatment with rFurin cleaved the RRRRR motif after the fifth or, rarely, the fourth arginine, as expected (Fig. S2F). By contrast, rKLK8 cleaved after the first, second, or third arginine. Control treatment with lysyl-endopeptidase (LE) that was used to activate rKLK8 did not on its own hydrolyze the peptide. These results suggest that KLK8 cleaves proActivin-A within the furin recognition motif. However by aligning this motif and its flanking sequences with two published consensus recognition motifs of KLK8, we could not predict which residues of proActivin-A are required for correct hemicleavage (Fig. 2D)[22]. Therefore, and since arginines at positions P1 and P4 typically suffice to enable cleavage by furin[23], we wondered if Activin-A hemicleavage by KLK8 can be inhibited by shortening the furin motif to only four arginines. To test this, we substituted arginine 310 at position P1 by alanine (R310A). Western blot analysis in SN of transfected FurKO#1 and DKO cells revealed that the R310A mutation indeed abolished the furin-independent conversion of A110 to A70, whereas in control B16-F1 cells, the remaining four arginines still enabled at least A70 formation by furin (Fig. 2E). To directly validate that one βA subunit was still cleaved by furin, we treated WT or this R310A mutant Activin-A with dithiothreitol (DTT). Analysis under these reducing conditions confirmed that while furin completely processed both βA subunits of wild-type Activin-A, the R310A mutant A70 generated by furin resolved into a mixture of uncleaved (56 kDa) and cleaved subunits (14 kDa) (Fig. S2G). Interestingly, correlating with this blockage of PCLP-mediated hemicleavage, the R310A mutant proActivin-A accumulating in FurKO#1 cell SN also failed to induce CAGA-Luc expression in reporter cells (Fig. 2F). By contrast, *Furin* wild-type B16-F1 cells still activated R310A mutant proActivin-A at least partially, allowing the CAGA-Luc to be induced 50-fold on average. Moreover, co-treatment of the reporter cells with CMK diminished the signaling activity of the R310A mutant by 43%, suggesting that R310A mutant Activin-A was partly activated by the signal-sending and partly by the recipient cells. Although a more quantitative comparison of the signaling strength of R310A versus wild-type Activin-A would require pure proteins, these results identify the R310A mutation as a useful tool to block Activin-A hemicleavage by PCLP activities but not by furin. In addition, they show that correlating with the inhibition of PCLP-mediated hemicleavage, the R310A mutation greatly increases the dependence of Activin-A on the coexpression of furin in *cis*.

To validate if R310 is needed for Activin-A hemicleavage by KLK8, the wild-type or R310A mutant precursors were each incubated with rKLK8 or rFurin. As shown in Fig. 2A (iii), the R310A mutant precursor resisted cleavage by KLK8 regardless of prior acidification. By contrast, incubation with rFurin still converted a significant fraction of the R310A mutant A110 to A70. These results confirm that the R310A mutation blocks the hemicleavage of Activin-A by KLK8, whereas furin-mediated cleavage is only partially inhibited, similar to what is observed in transfected cells. To investigate the underlying mechanism, we asked if KLK8 requires an arginine also at the analogous position of the RRRRR peptide. As shown in Fig. S2H, an alanine substitution at this position in a mutated RRRRA peptide shifted the rFurin-induced cleavage to the fourth arginine as expected. However, it did not prevent cleavage by rKLK8 after any of the first three arginines. These results suggest that the fifth arginine in the RRRRR sequence favors cleavage by KLK8 specifically in the conformational context of proActivin-A. To investigate how R310 enables Activin-A

hemicleavage by KLK8, we deleted this residue. The resulting ΔR310 mutant was processed similarly to wild-type Activin-A both in transfected FurKO#1 and control B16-F1 cells (Fig. 2G). Accordingly, a comparison of equal volumes of SNs of ΔR310 versus R310A or WT βA-transfected cells revealed that deletion of R310 also did not mimic the inhibitory effect of R310A on Activin-A signaling in CAGA-Luc reporter assays (Fig. 2H). Therefore, we hypothesized that the alanine substitution of R310 in proActivin-A blocks cleavage by KLK8 not by shortening the arginine cluster but by altering the flanking sequence. To test this, we flanked the new P1 arginine 309 of the R310A mutant furin motif with P2′ and P3′ residues identical to those of WT βA by replacing Gly111-Leu112 with Leu-Ala, giving rise to the sequence R309ALE (RALE) (Fig. 2D). As shown in Fig. 2G, H, the RALE mutant furin motif restored both Activin-A maturation in control B16-F1 cells and hemicleavage by PCLP activity in FurKO#1 cells, as well as signaling activity, even though it slightly reduced the electrophoretic mobility of cleaved A30. In good agreement, the RALE mutation also did not inhibit proActivin-A cleavage by rFurin in cell-free assays but rather increased it compared to wild-type (Fig. 2I). Importantly, the RALE mutant A110 was also converted to A70 similar to wild-type after in vitro incubation with rKLK8 (Fig. 2I). Overall, these results show that the conserved fifth arginine in the RRRRR motif is essential both for PCLP and furin activities to complement each other in cleaving proActivin-A in cells, and for hemicleavage by rKLK8 in vitro.

## KLK8 expression correlates with *INHBA* expression and with poor survival in multiple cancer types

To gain insight into KLK8 distribution across tumor types and normal tissues, we analyzed its mRNA and protein expression using public databases. In normal human tissues, KLK8 immunostaining was high in the skin, oral mucosa, and esophagus, followed by the vagina, cervix, and tonsil (Fig. S3A). In tumors, *KLK8* mRNA was particularly abundant in ovarian, head and neck, esophageal, and pancreatic cancers and to a lesser extent also in melanoma (Fig. 3A). The strongest *KLK8* upregulation relative to normal tissue was observed in pancreatic adenocarcinoma (159-fold), lung squamous cell carcinoma (59-fold), colon adenocarcinoma (35-fold), cervical cancers (23-fold), and in thymoma (3-fold) (Fig. S3B). Interestingly, Kaplan-Meier plots revealed that high *KLK8* expression is associated with shorter overall patient survival both in melanoma, and in 9 out of 12 other cancer types analyzed (Figs. 3B and S3C). Furthermore, at different stages of melanoma progression, tumors expressing *KLK8* were more frequent among patients at stage II, whereas no such enrichment was observed at more advanced stages (Fig. 3C). These data support a role for *KLK8* expression in cancer progression.

To evaluate a possible interaction of KLK8 with Activin-A in vivo, we assessed its correlation with *INHBA* expression in human melanoma. In primary melanoma, a trend for the expression levels of *KLK8* and *INHBA* to correlate did not reach significance (Fig. 3D). We also analyzed *KLK8* expression in a panel of 13 human melanoma cell lines. While *KLK8* mRNA was detected in six of the cell lines examined, four of these tested negative for *INHBA* mRNA (Fig. 3E). Interestingly, however, stable transduction of YUMM3.3 cells with *INHBA* seemed to increase *Klk8* expression (Fig. S1J). To evaluate a causal relationship, we exposed YUMM3.3 cells to acute *INHBA* overexpression with or without treatment with SB 431542, or to increasing concentrations of recombinant Activin-A protein. *Klk8* mRNA increased on average 15-fold within 48 hrs of βA transduction, whereas SB treatment significantly reduced *Klk8* expression to levels comparable to those in YUMM3.3 cells without βA. Moreover, treatment with recombinant Activin-A showed a similar trend (Fig. 3F). Treatment with Activin-A also increased the *KLK8* mRNA levels in human C81-61 melanoma cells, whereas a similar trend did not reach significance in SH-4 cells (Fig. 3G). Besides revealing significant *KLK8* expression in skin and in multiple cancer types, including melanoma, these results indicate

that a potential role for Activin-A signaling in promoting *KLK8* expression is unlikely a common feature among skin melanoma at all tumor stages.

## Klk8 stimulates B16-F1 cell proliferation and anchorage-independent growth in vitro

To assess the impact of *Klk8* expression on cancer cells, we asked if its depletion by RNAi alters the proliferation of B16F1-Ctrl or -βA cells in 2D cultures, or their anchorage-independent survival in soft agar. Transfection with *Klk8* siRNA led to a 5-fold increase in dead cells 72 h after transfection (Fig. S4A). While this increase in cell death was deemed suitable for in vitro assays, we reasoned that the loss of a fraction of cells prior to tumor grafting would bias the outcome. Therefore, as an alternative system to acutely deplete Klk8, we generated IPTG-inducible shKlk8 cell lines. Analysis by RNAscope and RT-qPCR showed that two of four IPTG-inducible *Klk8* shRNAs (shKlk8) significantly depleted *Klk8* mRNA in B16F1-βA cells as compared to control shRNA (shLuc) (Fig. S4B-D). In B16F1-βA cells, induction of either of these two shRNAs by IPTG treatment within 72 hrs significantly diminished the number of viable cells detected by AlamarBlue staining relative to untreated or shLuc expressing cells (Fig. 4A). However, in the B16F1-Ctrl cells, a similar effect only reached significance for one of the two *Klk8* shRNAs, and only when compared to the corresponding IPTG-treated controls with and without shLuc (Fig. 4B). These data suggest that IPTG treatment alone was not completely neutral. Prompted by these observations, we decided to also test the effect of *Klk8* depletion on cell survival and non-adherent growth during colony formation in soft agar. While IPTG treatment alone did not significantly alter the number of colonies formed by B16F1-Ctrl and -βA cells, shKlk8 sequences diminished it relative to both untreated or shLuc expressing controls (Fig. 4C). These results suggest that even though IPTG treatment alone can boost B16F1-Ctrl cell proliferation non-specifically at least in 2D cultures, IPTG-induced *Klk8* knockdown significantly diminishes the growth of B16F1-Ctrl and -βA cells both in adherent and non-adherent culture conditions, regardless of the βA transgene.

## Inducible *Klk8* RNAi diminishes the growth of tumors specifically if they secrete Activin-A

To evaluate the influence of *Klk8* knockdown on Activin-A processing and signaling in vivo, syngeneic B16F1-Ctrl or -βA melanoma grafts were induced to express shKlk8 or shLuc control by administering IPTG to the drinking water of tumor-bearing hosts (Fig. 5A). Unexpectedly, compared to controls without IPTG, IPTG administration tended to markedly accelerate tumor growth of B16F1-Ctrl grafts even in the absence of any shRNA (Fig. 5B, top), whereas in βA-expressing tumors that were grafted in parallel, we observed an opposite trend, with IPTG tending to slow tumor growth rather than facilitating it (Fig. 5B, bottom). Since IPTG thus masked the tumor growth advantage conferred by βA, this allolactose-mimicking reagent is apparently not neutral. Importantly, induction of either of the two *Klk8* shRNAs by IPTG nevertheless further diminished tumor growth specifically in B16F1-βA, but not in B16F1-Ctrl tumors (Fig. 5B). Induction of shLuc had no such effect, indicating specificity. These results support a role for Klk8 in potentiating Activin-A induced tumor growth.

## Klk8-mediated proActivin-A cleavage can be obscured by furin also in tumors

Since the contribution of Klk8 to Activin-A processing in vitro can be obscured by the overlapping activity of furin, we asked if this is also true in vivo. Analysis of tumor extracts and of plasma from tumor-bearing mice revealed that *Klk8* knockdown did not overtly decrease Activin-A processing in furin wild-type tumors, or the accumulation of mature Activin-A in plasma of tumor-bearing hosts (Fig. 5C, D). To directly test if Klk8 and furin activities complement each other in vivo, we expressed the IPTG-inducible control shLuc or *Klk8* shRNAs in B16F1 FurKO-βA

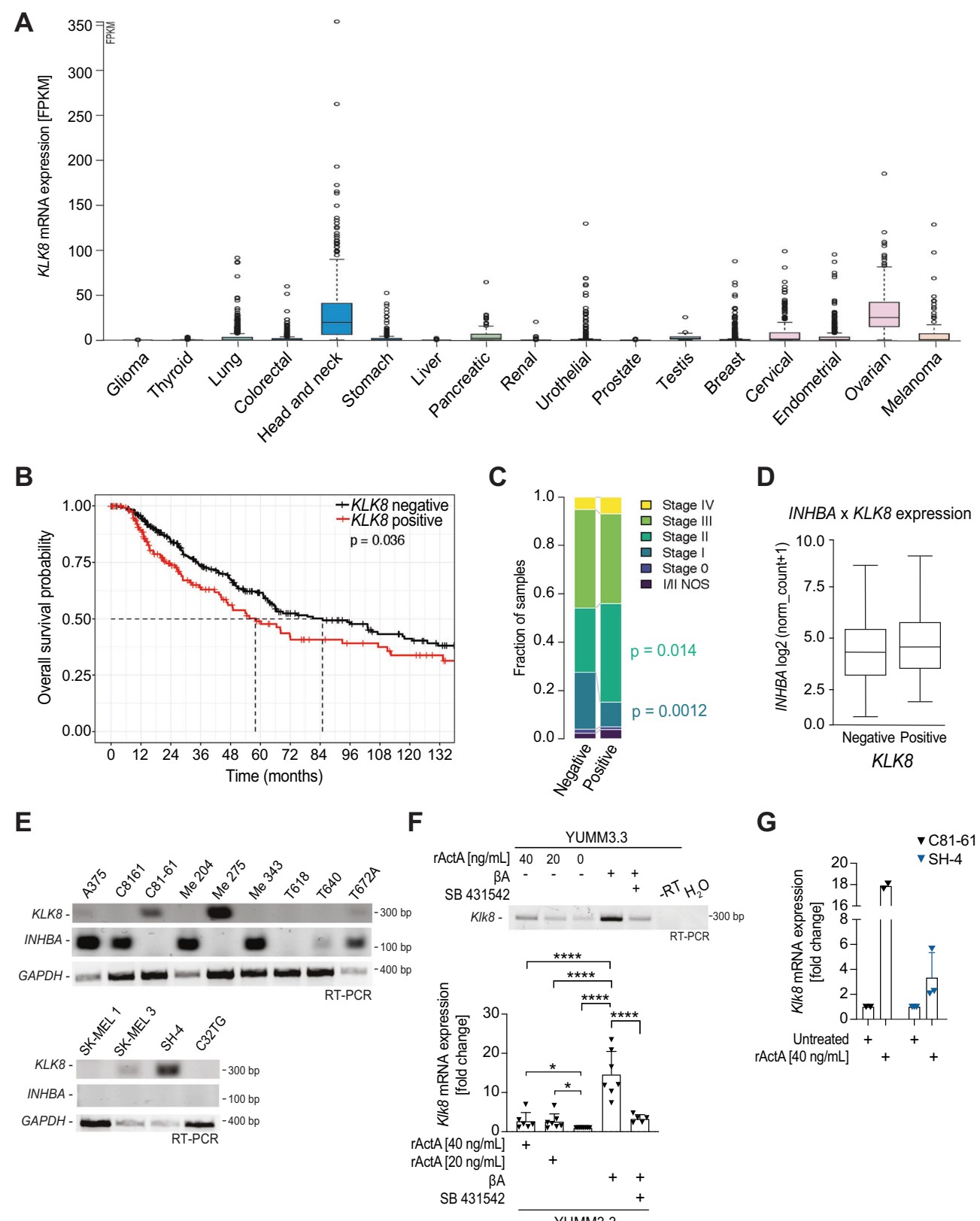

melanoma grafts lacking furin. Induction of shKlk8_4 expression significantly decreased the tumor growth compared to shLuc control tumors, and a similar strong trend was seen in tumors expressing shKlk8_1 (Fig. 5E). Furthermore, Western blot analysis revealed that compared to shLuc control, both *Klk8* shRNAs significantly enriched the tumor lysates for uncleaved A110 relative to A30 at least by the time the

tumors reached the endpoint (Fig. 5F, G). The ratio of the processing intermediate A70 relative to A30 tended to increase alongside. However, *Klk8* knockdown did not significantly change the levels of A30 in the circulation compared to control groups with shLuc or with IPTG treatment alone (Fig. 5F, G). Taken together, these results strongly support a role for Klk8 in promoting melanoma growth by contributing

**Fig. 3 | Regulation of KLK8 expression in human cancers and in melanoma cell lines. A** *KLK8* mRNA expression in human cancers according to The Human Protein Atlas. **B** Kaplan Meier plot showing the overall survival of *KLK8* mRNA expression in melanoma patients (primary tumors + metastatic). Patients were ranked as negative when no *KLK8* expression was detected, and as positive when *KLK8* expression was > 0. (n = 445; 170 KLK8-positive, and 275 KLK8-negative, Logrank test). **C** *KLK8* expression across melanoma stages (primary tumors + metastatic patients with pathological status). NOS, Not Otherwise Specified. (n = 410, 159 KLK8-positive, and 251 KLK8-negative, Two-sided empirical Bayes quasi-likelihood F-test). **D** Lack of significant correlation between *INHBA* and *KLK8* mRNA expression levels in primary tumors of melanoma patients (n = 103; 70 KLK8-positive, and 33 KLK8-negative). **E** RT-PCR analysis of *KLK8* and *INHBA* mRNAs in a

panel of 13 human melanoma cell lines. Data represent one of three independent experiments with similar results. **F** RT-PCR (top) and RT-qPCR (bottom) analyses of endogenous *Klk8* relative to *Gapdh* mRNA expression in YUMM3.3 cells treated with 20 or 40 ng/mL of rActA for 48 hrs, or transduced with βA with or without treatment with 10 μM SB 431542 for 48 hrs. Relative *Klk8* expression is normalized to the expression of *Klk8* in control YUMM3.3 cells and shows the average of at least five experiments ± SD (One-way ANOVA). **G** RT-qPCR analysis of endogenous *KLK8* relative to *Gapdh* mRNA expression in C81-61 and SH-4 cells with or without treatment with 40 ng/mL of recombinant Activin-A. Relative *KLK8* expression is normalized to the expression of *KLK8* in untreated cells. Data show the average of two to three experiments ± SD. *$p < 0.05$, ****$p < 0.0001$.

to Activin-A precursor processing, even though this contribution can be obscured by the overlapping activity of furin.

### The KLK8-resistant R310A mutant Activin-A is not sufficiently activated by furin alone to stimulate tumor growth

An alternative interpretation is that Klk8 cleavage of some other substrate(s) potentiated the function of Activin-A indirectly. To selectively block the direct effect of Klk8 on Activin-A, we analyzed B16-F1 melanoma grafts that secrete the KLK8-resistant proActivin-A encoded by R310A mutant βA. Since the R310A mutant is still significantly cleaved by furin and competent to signal, at least in vitro, we compared its function with that of mS1 mutant Activin-A precursor where the penta-arginine motif of site 1 was substituted by five alanines (Fig. 6A). Only wild-type βA, but not R310A or mS1 increased tumor growth compared to Ctrl (Fig. 6B). Western blots of tumor extracts confirmed the presence of mature Activin-A (A30) in WT-βA tumors, whereas mS1 accumulated as an uncleaved precursor. Importantly, R310A-βA still formed A70, but no detectable A30 (Fig. 6C). Furthermore, correlating with this inhibition of A30 formation, mS1 and R310A mutant βA also failed to decrease the body or heart weights of the tumor-bearing hosts (Fig. 6D). We therefore asked if processing also regulates the release of Activin-A into the circulation. Analysis of plasma by ELISA and immunoblotting showed that WT-βA tumors increase circulating Activin-A in plasma of tumor-bearing mice more than 30-fold above baseline to around 35 ng/ml[10], which is still below the levels of endogenous Activin-A released into the plasma by granulosa cell-derived tumors of *Inha* knockout mice[24]. By contrast, circulating mS1 and R310A Activin-A mutants were below detection (Fig. 6E, F). Taken together, these results show that the R310A mutation inhibits both the Activin-A induced tumor growth advantage as well as the release of Activin-A into the circulation.

### Inducible Klk8 RNAi or blockade of Klk8 cleavage by the R310A mutation in βA-expressing tumors enriches infiltrating CD8⁺ T cells.

To evaluate possible mechanisms of how Activin-A cleavage by Klk8 facilitated tumor growth, we first analyzed its impact on tumor-infiltrating CD8⁺ T cells. A comparison of the above tumors at the endpoint revealed that the average frequency of infiltrating CD8⁺ T cells was decreased in wild-type βA compared to Ctrl tumors as expected[10], but not in tumors expressing R310A mutant βA (Fig. S5A), suggesting that partial processing of the Klk8-resistant R310A mutant proActivin-A is not sufficient to deplete these T cells. The proportion of CD8⁺ T cells in B16F1-βA melanoma grafts also increased if they expressed *Klk8* shRNA, relative to shKlk8-expressing B16F1-Ctrl tumors, and a similar trend was seen relative to βA tumors expressing shLuc or no shRNA (Fig. S5B). These data suggest that shKlk8 specifically alters the function of Activin-A. Indeed, no significant differences in the proportion of infiltrating CD8⁺ T cells were observed among any other IPTG-treated βA versus Ctrl groups in either the presence or absence of shLuc, in line with our aforementioned observation that treatment with IPTG alone also blunted the βA-induced tumor growth advantage. Thus, we conclude that while

Activin-A diminishes CD8⁺ T cell infiltration in tumors that express Klk8 and furin, Klk8-independent cleavage alone can stimulate it. To determine if *Klk8* RNAi led to an increase in apoptotic cells, histological sections of βA-expressing tumors were stained with antibodies against cleaved caspase 3. *Klk8* depletion significantly increased cleaved caspase 3 staining (Fig. S5C), whereas a trend to reduce staining of the cell proliferation marker Ki-67 did not reach significance (Fig. S5D). Importantly, Western blot analysis of tumor extracts showed that *Klk8* knockdown also significantly decreased the levels of pSmad2 relative to total Smad2 (Fig. S5E). These data further corroborate our interpretation that precursor processing by Klk8 facilitated tumor growth by specifically potentiating Activin-A signaling.

## Discussion

ProActivin-A can be partially cleaved independently of furin and other known basic residue-specific PCs to generate the hemicleaved processing intermediate A70, but the regulation and function of hemicleavage remained elusive[15]. Here, an RNAi screen for proteases mediating A70 formation in *Furin* knockout B16-F1 melanoma cells identified Klk8. This trypsin-like protease cleaved the penta-arginine furin recognition motif of proActivin-A in cell-free assays and in a synthetic peptide, as expected of the PC-like protease. Mutating the fifth arginine of this motif to alanine abolished hemicleavage by KLK8 and the tumor growth advantage of syngeneic B16-F1 melanoma grafts conferred by Activin-A. Importantly, the growth of B16-F1 melanoma secreting Activin-A also decreased upon depletion of *Klk8*, despite the overlapping activity of furin. Human *KLK8* expression correlated with poor patient outcomes in many cancer types, including melanoma, suggesting that inhibition of KLK8 holds promise as a future therapeutic strategy.

### RNAi screening for candidate PC-like protease activities in melanoma cells

Since furin cleaves a plethora of secreted proteins essential for human health, we asked if Activin-A can be targeted by inhibiting other proteases. Here, RT-qPCR analysis of *Pcsk* genes encoding basic residue-specific PCs confirmed that B16-F1 cells only express furin/*Pcsk3* and PC7/*Pcsk7*[15]. PC7 activity in these cells contributes to Notch1 and ADAM10 processing, but not to furin-independent A70 formation[15]. B16-F1 cells also expressed PCSK9. PCSK9 cleaves after non-basic amino acids[14], and its depletion had no overt effect on Activin-A. Our siRNA library screen for proteases enabling furin-independent Activin-A signaling in co-cultured SMAD3 reporter cells instead identified Klk8 and Adamts10 as top candidates. Human ADAMTS10 is not expressed in melanoma in the TCGA database, and it is not commercially available in recombinant form. However, it is mutated in Weill-Marchesani syndrome, a connective tissue disorder linked to defects in the processing of fibrillin-1 and -2 that perturb the architecture of microfibrils and their ability to store latent TGF-β and several BMPs[25–27]. In addition, Adamts10 has been shown to stimulate TGF-β signaling in retinal ganglion cells of developing zebrafish[28]. Of

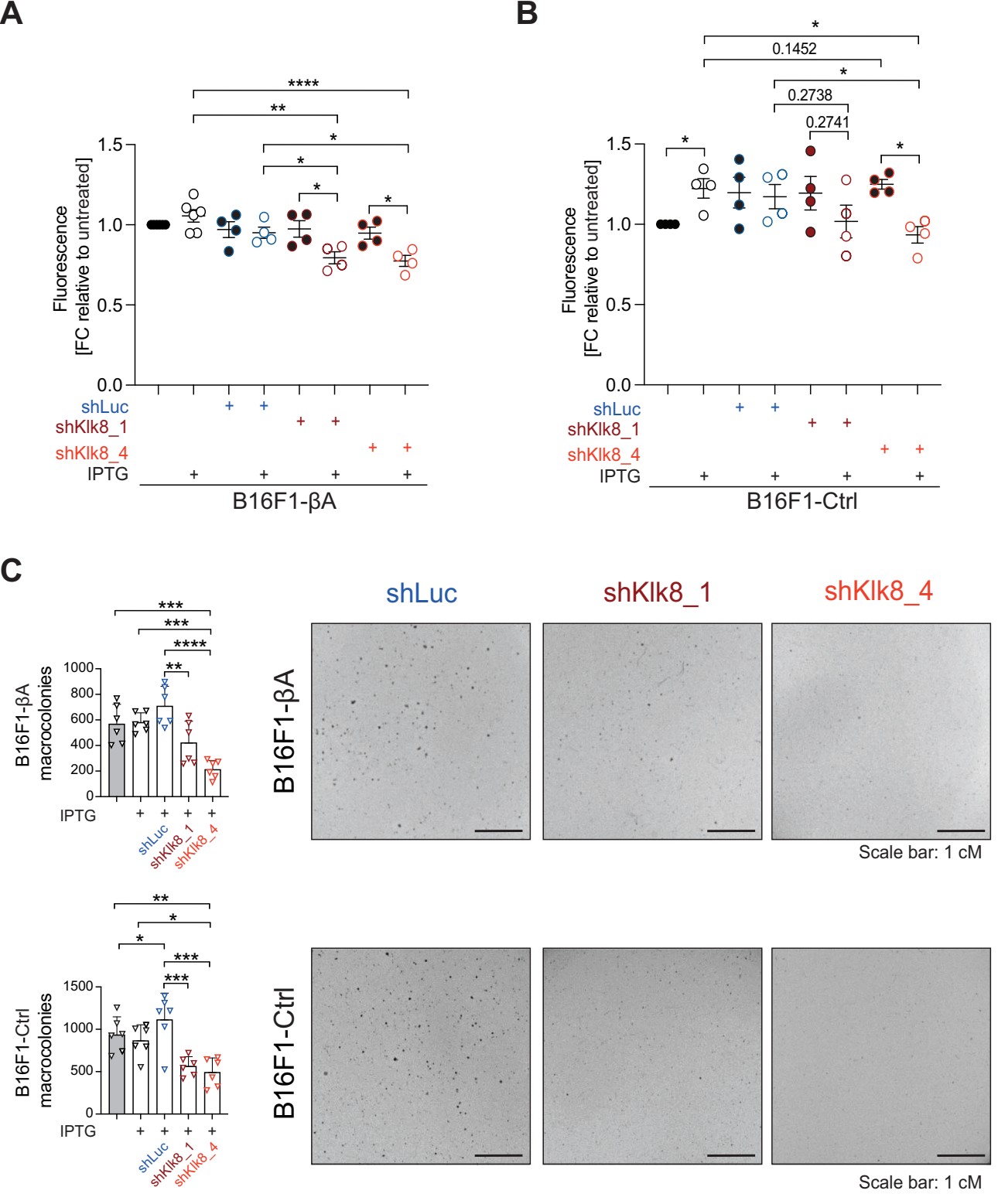

**Fig. 4 | IPTG-inducible Klk8 knockdown diminishes B16F1-Ctrl and -βA cell proliferation and anchorage-independent colony formation in soft agar.**
**A** Alamar Blue staining of viable B16F1-βA cell lines after treatment with empty vehicle or 100 μM IPTG for 72 hrs to induce control shRNA (shLuc), shKlk8_1, or shKlk8_4. Data show the average of four to six experiments ± SD (Two-sided Welch's *t*-test). **B** As in (**A**) but in B16F1-Ctrl cells. Data show the average of four experiments ± SD (Two-sided Welch's *t*-test). **C** Macrocolonies formed within 14 d in soft agar by B16F1-βA (top) or B16F1-Ctrl cells (bottom) treated or not with 100 μM IPTG, and by B16F1-Ctrl or -βA shLuc, shKlk8_1, or shKlk8_4 cells treated with 100 μM IPTG. Data represent two experiments with three replicates ± SD (One-way ANOVA). Representative crystal violet stainings of cells at the experimental endpoint are shown to the right. *$p < 0.05$, **$p < 0.01$, ***$p < 0.001$, ****$p < 0.0001$.

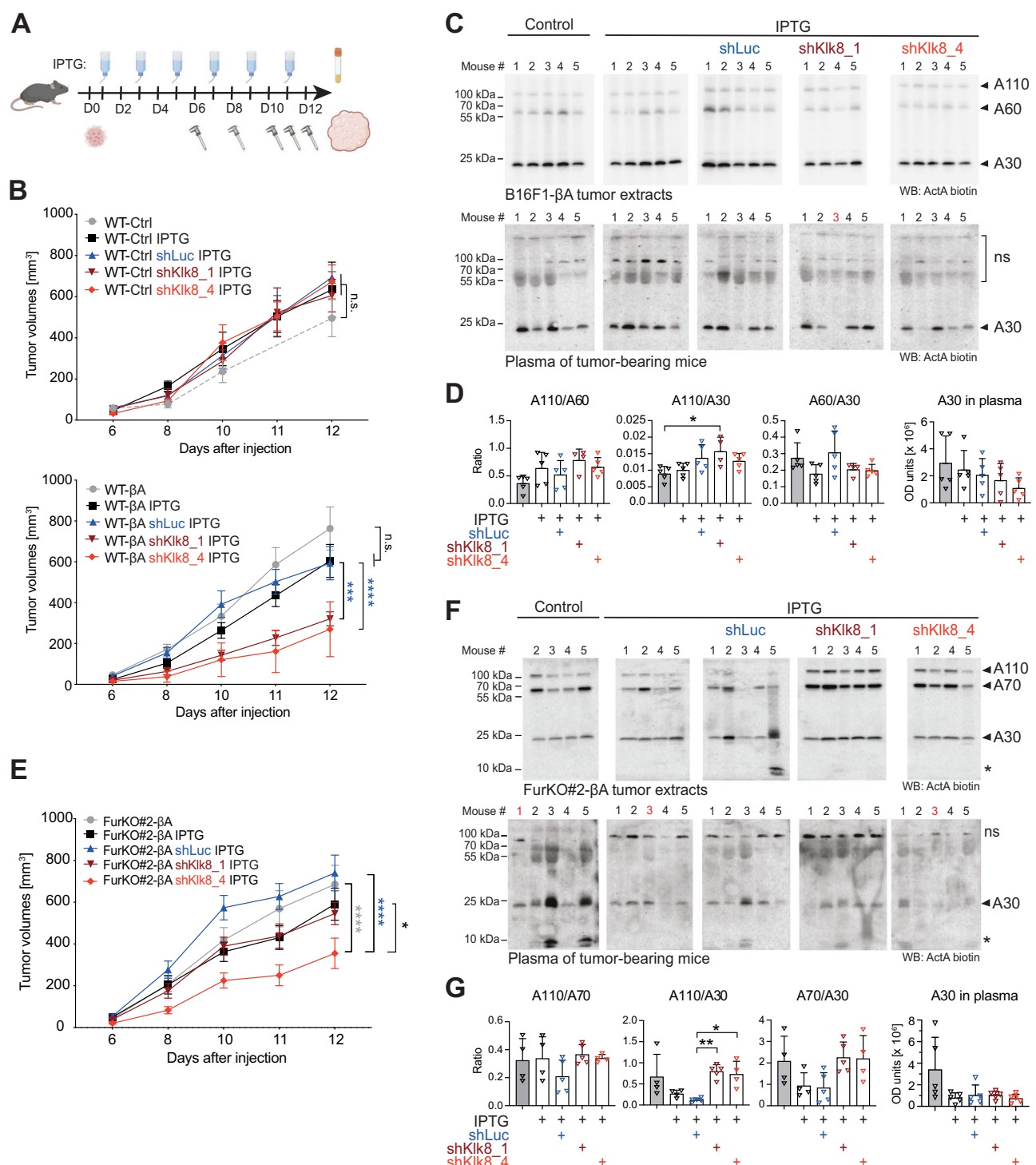

**Fig. 5 | Klk8 depletion in syngenic B16-F1 melanoma grafts spefically slows the growth of INHBA-expressing tumors, but its influence on Activin-A processing can be obscured by furin. A** Procedure to monitor the influence of IPTG-inducible *Klk8* depletion on B16-F1 melanoma grafts. Created in BioRender. Bulliard, M. (2025) https://BioRender.com/k13i682. **B** Growth curves of *Furin* wild-type B16F1-Ctrl (WT-Ctrl) (top) or B16F1-βA (WT-βA) cells (bottom) with or without shLuc, shKlk8_1, or shKlk8_4 after intradermal grafting in syngeneic C57BL/6 J mice. Data show the average of 9 to 10 mice per group ± SEM (Two-way ANOVA). n.s., not significant. **C** Activin-A Western blots of extracts of B16F1-βA tumors (top) or plasma (bottom) from the cohort of tumor-bearing mice in (**B**) at the experimental endpoint. ns, non-specific. Note that in mouse #3 (highlighted in red) of the shKlk8_1 group, the tumor remained too small for Western blot analysis, and

Activin-A in plasma was below detection. **D** Optical density (OD) values ± SD (One-way ANOVA) of the adjusted volumes of protein bands corresponding to the indicated forms of Activin-A in the tumor extracts and plasma samples shown in (**C**). Data represent at least four tumor-bearing mice per genotype. **E** Growth curves of B16F1 FurKO#2-βA tumor grafts without or with shLuc, shKlk8_1, or shKlk8_4 in C57BL/6 J syngeneic mice. Data show the average of 9 to 10 mice per group ± SEM (Two-way ANOVA). **F** Activin-A Western blots of extracts of the tumors in (**E**), and in plasma of the tumor-bearing mice at the experimental endpoint. Asterisk, possible degradation product. ns, non-specific. Red numbers, as in (**C**). **G** As in (**D**), but for the tumor extracts and plasma samples shown in (**F**). Data represent at least four tumor-bearing mice per genotype. *$p < 0.05$, **$p < 0.01$, ***$p < 0.001$, ****$p < 0.0001$.

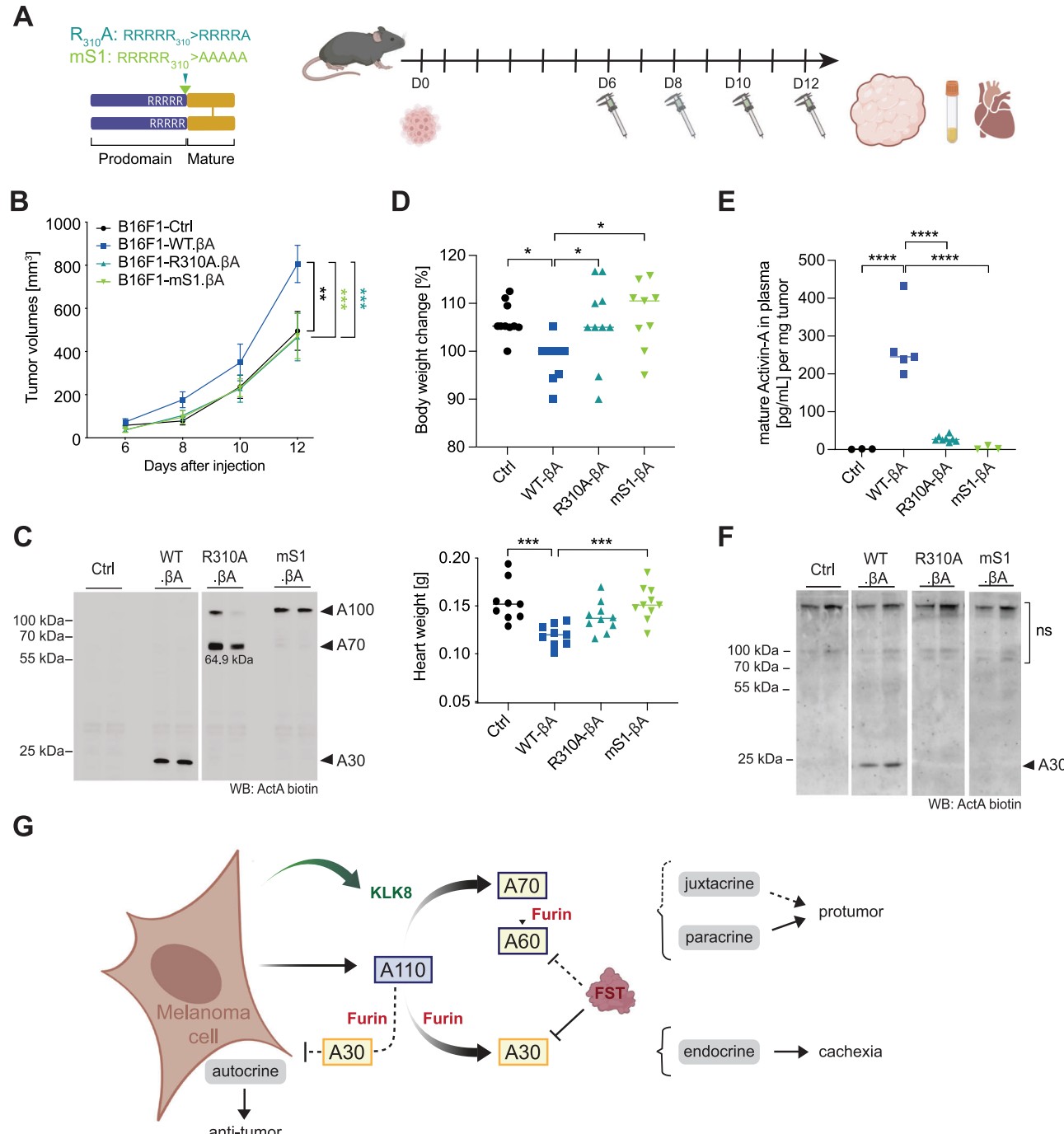

**Fig. 6 | The R310A mutation that blocks Klk8-mediated hemicleavage of Activin-A abrogates its tumor-promoting activity. A** Schematic of the mutations in proActivin-A used to block hemicleavage by KLK8 (R310A), or to inhibit both KLK8 and furin (mS1) (left), and of the strategy to analyze their influence on Activin-A induced tumor growth (right). Created in BioRender. Bulliard, M. (2025) https://BioRender.com/f04n549. **B** Growth curves of syngeneic B16-F1 melanoma grafts expressing control lentivirus (Ctrl), or wild-type (WT) βA, or the R310A, or mS1 cleavage site mutant derivatives. Data show the average of ten mice per group ± SEM (Two-way ANOVA). **C** Activin-A Western blot of 2 representative tumor lysates per genotype at the experimental endpoint. **D** Changes in body (top) and heart weights (bottom) between experimental start- and endpoints. Data show the average of ten mice per group (One-way ANOVA). **E** Mature Activin-A detected by ELISA in plasma at the experimental endpoint. Data show the amount of mature Activin-A in pg/mL per mg of tumor in 3 to 5 tumors per genotype (One-way ANOVA). **F** Western blot analysis confirming the depletion of Activin-A in plasma by both the mS1 and R310A mutations. Two tumor-bearing mice were analyzed per genotype. **G** Schematic of Activin-A processing in the absence of KLK8 and its effect on tumor growth. Created in BioRender. Bulliard, M. (2025) https://BioRender.com/ w96o575. *$p < 0.05$, **$p < 0.01$, ***$p < 0.001$, ****$p < 0.0001$.

note, both Adamts10 and its substrate fibrillin-1 also contain essential furin cleavage sites[25,29]. Therefore, future studies should investigate if furin regulates Activin-A processing in part indirectly via an Adamts10/fibrillin axis.

Additional candidate regulators of Activin-A signaling emerging from our screen included the complement C1r subcomponent-like (C1rl), plasminogen (Plg), and Klk6. C1rl appears to function as a protease in the ER, but a mechanism to activate the zymogenic form is

lacking[30,31]. By contrast, Plg can be converted to plasmin to then activate various growth factors, in part by degrading extracellular matrix[32]. *Plg* RNAi did not impair hemicleavage, suggesting that it inhibited Activin-A signaling indirectly. Significant hemicleavage also persisted after RNAi of *Klk6*, and KLK6 did not process proActivin-A in cell-free assays. However, its expression correlates with that of *KLK8* in ovarian, pancreatic, and colorectal cancer[33–35]. Whether and how KLK6 stimulates activin signaling in tumors or in normal tissues remains to be investigated.

## KLK8 can convert proActivin-A to hemicleaved form

Analysis of the influence of Activin-A processing on SMAD3 signaling in a reporter cell line after *Klk8* knockdown in furin-deficient B16-F1 melanoma cells revealed that this kallikrein converts proActivin-A to A70 independently of furin. In a separate study, we recently found that cysteines C35 and C38 in the Activin-A prodomain form allosteric disulfide bonds with C314 and C322 that are essential to prevent precocious access of furin to the nearby penta-arginine motif before secretion, but that one βA subunit can bypass this inhibition, likely by interacting with a rate-limiting endogenous protein that remains to be identified[16]. Interestingly, incubation with rKLK8 revealed that direct cleavage of proActivin-A required prior transient acidification, indicating additional layers of regulation. By default, we acidified samples during 1 hr to pH 2, but exposure to pH 6.5 already enabled significant cleavage. Changes of the pH can dissociate salt bridges and hydrophobic and aromatic/proline interactions in various proteins[36]. For example, even though enzymatic KLK8 activity is optimal between pH 7 to 9[37], a low pH in vivo may facilitate its activation by KLK5, as shown for the zymogenic maturation of KLK7[38]. Acidification releases several kallikreins from inhibitory proteins, including KLK5, KLK7, and KLK14, raising the possibility of a similar effect on KLK8[39]. However, in our setting, we transiently acidified Activin-A, not KLK8. Given the adverse effect on cleavage by rFurin, lowering the pH may alter the conformation of proActivin-A itself or its association with an interacting factor. In keeping with this idea, acidification can dislodge TGF-β from a latent complex with its prodomain[40,41]. Furthermore, furin cleavage of Notch can be inhibited by the specific interacting factor Botch[42]. Interestingly, keratinocytes maintain skin barrier functions by producing epidermal lamellar bodies that acidify the extracellular space below pH 5.5. Their cargo includes KLK8[43]. Lowering the extracellular pH below 6.5 in turn may increase KLK8 expression and secretion, as shown in gastric cancer cells[44]. These observations are consistent with the idea that acidification may be a physiological stimulus of Activin-A hemicleavage. Moreover, incubation with KLK5 mimicked the ability of KLK8 to generate A70 in SNs containing A110, suggesting possible functional overlap in cells or tumors expressing both of these proteases, and/or that KLK5 might activate endogenous proKLK8. During liver regeneration after surgery, a latent complex of mature TGF-β1 at the surface of hepatic stellate cells is activated by KLK3[45]. However, unlike the furin-independent hemicleavage by KLK8 described here for Activin-A, KLK3 activates latent TGF-β by degrading a dimer of the prodomain after its prior cleavage by furin[46].

## Role of arginine 310 in proActivin-A processing by KLK8 versus furin in vitro

Previous analysis of other substrates established that KLK8 can cleave a relatively broad range of sequences containing Arg or Lys at position P1[47]. While the furin motif of Activin-A satisfies this criteria, cleavage of KLK8 substrate also depends on their charge and on the orientation of the scissile bonds, and some amino acids are favored over others in the flanking sequences[22,37,48]. Here, a synthetic peptide comprising the penta-arginine motif of proActivin-A was cleaved by KLK8 after the first, second, or third arginine, but not after the fourth or fifth, concurring with published KLK8 consensus motifs where arginines are among the favored residues at positions P2', and P1 to

P3, but not in P4 or P5 (Fig. 2D)[22]. Interestingly, alanine substitution of the fifth arginine in this peptide only seemed to shift the main scissile bond to the first arginine, whereas in full-length R310A mutant proActivin-A, it completely abolished KLK8 cleavage. The R310A mutation also diminished cleavage by furin specifically in full-length proActivin-A. Importantly, however, both the formation of hemicleaved A70 by KLK8, as well as its further maturation by furin, were fully rescued if $R_{310}$ was simply deleted, or if the $RA_{310}GLE$ sequence in R310A mutant proActivin-A was mutated to $RA_{310}LE$. Taken together, these findings strongly support our interpretation that conformational constraints regulate the positioning of scissile bonds in the penta-arginine motif of full-length proActivin-A, but not in the identical sequence of a flexible peptide.

Our rationale for generating the RALE mutant sequence was that an engineered Leu-Glu dipeptide at the P2'-P3' position of the furin recognition motif in the related Nodal precursor increases its processing and signaling[49,50]. Here, Leu-Glu at the corresponding positions relative to R309 in the RALE mutant construct fully rescued both Activin-A processing and signaling. Interestingly, however, the electrophoretic mobility of this mutant mature Activin-A was shifted by 2.6 kDa. A similar mobility shift is seen if cysteine(s) C314 or C322 are substituted by alanine[51], suggesting that the RALE sequence may disrupt the formation of a disulfide bridge by these cysteines, thereby increasing the freedom of the nearby multibasic motif and hence the flexibility of how it can be cleaved.

## Regulation of Activin-A processing by arginine 310 and by Klk8 in vivo and its impact on tumor growth

We found that IPTG-inducible *Klk8* shRNAs specifically impaired the growth of B16F1-βA melanoma grafts, but not of B16F1-Ctrl tumors. This observation strongly supports a role for Klk8 in directly regulating the bioavailability of Activin-A. One caveat is that independently of its effect on shRNAs, IPTG treatment blunted both the CD8+ T cell exclusion and the tumor growth conferred by the βA transgene in this and other melanoma models[9,10]. Nevertheless, *Klk8* shRNAs slowed the growth of βA tumor grafts below that of Ctrl tumors. This finding suggests that Klk8 primarily diminishes tumor-suppressive Activin-A signaling, possibly without increasing its tolerogenic pro-tumor function. In keeping with this interpretation, induction of *Klk8* shRNAs by IPTG enriched the frequency of CD8+ T cells only in βA compared to Ctrl tumors, but not when compared to βA tumors expressing control shRNA. In addition, it increased apoptotic cells marked by cCasp3 staining, an effect similar to the one of sustained cell-autonomous activin receptor activation within the melanoma cell themselves[9]. By contrast, cell proliferation in vivo was unchanged, even though shKlk8 induction decreased the proliferation and anchorage-independent growth in vitro. Thus, besides cleaving proActivin-A, Klk8 likely has an additional function in enhancing cell survival which in vivo could be obscured by compensatory signals. Further testing of how *Klk8* knockdown enabled Activin-A to reduce tumor growth will require prior identification of the cells mediating pro- and/or anti-tumor effects of Activin-A on immune infiltrates, and of activin-specific target genes to reliably quantify potential changes in signaling dynamics.

Analysis in tumors lacking *Furin* showed that *Klk8* knockdown stabilized proActivin-A (A110) and hemicleaved A70 without changing the levels of mature form (A30). This suggests that Klk8 promotes tumor growth by locally regulating the flux of differentially processed forms of Activin-A, rather than by increasing the steady-state level of fully mature form. Importantly, A110 and A70 similarly accumulated in tumors expressing R310A mutant proActivin-A. Although one βA subunit in this mutant can still be cleaved by furin to generate A70 and induce significant SMAD3 signaling when secreted by *Furin* wild-type cells in vitro, it failed to mediate a tumor growth advantage or disadvantage in vivo. Thus, hemicleaved Activin-A clearly neither

promotes nor inhibits tumor growth without undergoing further maturation. Based on these observations, we propose that hemi-cleavage by Klk8 hinders tumor-suppressive Activin-A signaling by interfering with the stepwise proteolytic maturation mediated by furin. In this model, Klk8 may facilitate the attenuation of tumor-suppressive signaling within the cancer cells[9] or diminish the bioavailability of Activin-A for another specific cell type that facilitates their killing (Fig. 6G). Regardless of the mechanism, our findings identify KLK8 as an attractive strategy to block the stimulation of tumor growth by Activin-A, at least in melanoma.

## Expression of KLK8 in human skin and during melanoma progression

A comparison of KLK8 distribution across tissues previously revealed the highest expression in the CNS, skin, and in ovaries[52]. *Klk8* expression in the skin is upregulated during inflammation and important to promote wound healing[53,54], and to prevent hyperkeratosis in a psoriasis model, likely by cleaving desmosomal cell-cell adhesion proteins[55]. Whereas our analysis of public databases confirmed high expression of *KLK8* in skin keratinocytes, KLK8 is not secreted by healthy melanocytes or dermal fibroblasts[37,56]. However, a majority of human and murine melanoma cell lines analyzed here expressed *KLK8* mRNA. In at least two of these, *KLK8* expression increased in the presence of Activin-A, even though *KLK8* and *INHBA* were not consistently co-expressed in the human melanoma cohort examined here. Our analysis of public datasets confirmed that high *KLK8* expression correlates with poor patient outcomes in 10 out of 13 human cancer types examined, as described previously for pancreatic, endometrial, and breast cancers, as well as for clear cell renal cell carcinoma and urothelial bladder carcinoma[57–62]. In melanoma, *KLK8* expression peaked during the transition of primary tumor to metastatic disease. This result agrees with published data showing transient upregulation of *KLK8* expression at the T3 stage preceding the establishment of distant metastasis[63]. Recent studies also identified KLK8 among useful markers to predict cancer progression and immune response and to classify melanoma into subgroups to select distinct treatments[64,65]. These observations further increase the attractiveness of KLK8 as a potential target for future cancer therapies in melanoma and beyond.

# Methods

## Study approval

This research complied with all relevant ethical regulations. All protocols for animal studies were in accordance with 3 R guidelines and with the animal welfare regulations of the Federal Food Safety and Veterinary Office of Switzerland and approved by the ethics committee and the veterinary administration of the canton of Vaud (license VD2774.2).

## Cell lines

All complete cell culture media were supplemented with 1% GlutaMAX (Gibco), 50 μg/mL Gentamicin (Gibco), and 10% fetal bovine serum (or 15% in the case of McCoy's 5 A medium). HepG2 (cat. no. HB-8065), HEK293T (cat. no. CRL-3216), B16-F1 (cat. no. CRL-6323), SK-MEL-1 (cat. no. HTB-67), SK-MEL-3 (cat. no. HTB-69), SH-4 (cat. no. CRL-7724), and C32TG cells (cat. no. CRL-1579) were purchased from the American Type Culture Collection (ATCC). HEK293T, HepG2, human SH-4, and mouse B16-F1 melanoma cell lines were maintained in DMEM (Gibco, ThermoFisher Scientific, Waltham, MA, USA). C32TG and the non-adherent SK-MEL-1 cells were maintained in EMEM (Gibco), and SK-MEL-3 in McCoy's 5a (Gibco). EMEM for the human C32TG melanoma cell line was supplemented with 50 μM of 6-thioguanine (Sigma-Aldrich, Missouri, USA). Human Me 204, Me 275, Me 343, and T672A cell lines from metastases of cutaneous melanoma patients were derived at the Ludwig Institute for Cancer Research (CHUV, Lausanne, CH) and maintained in RPMI (Gibco). YUMM3.3 cells were provided by

Dr. Anna Obenauf (IMP, Vienna, AU) and maintained in DMEM/F12 (Gibco). The A375 cell line was provided by Dr. Mélanie Tichet (Hanahan lab, EPFL) and maintained in RPMI (Gibco). C8161 cells were provided by Dr. Mary Hendrix (Shepherd University, Shepherdstown, USA). Human SCC-12 and SCC-13 squamous cell carcinoma cells (provided by Dr. Freddy Radtke, EPFL) were maintained in keratinocyte Serum-Free Medium (SFM) (Gibco, 17005042) supplemented with 25 μg/mL Bovine Pituitary Extract (BPE) (Gibco, 13028014) and 0.2 ng/mL human EGF (Gibco, PHG0311). SSC-12, and SCC-13 were detached using accutase (CELLnTECH) 1x for 5 min at 37 °C. All other adherent cell lines were detached using Trypsin-EDTA 1x (Biowest, Nuaillé, FR). SK-MEL-1 are non-adherent cells. They were collected by centrifugation at 1000 x g for 5 min. Three B16F1 FurKO clones and one DKO CRISPR clone were generated by CRISPR/Cas9 editing single guide RNAs[15]. B16-F1, FurKO, DKO, and YUMM3.3 cells that stably overexpress myc-tagged INHβA (βA) or an empty vector Ctrl were previously generated by stable transduction with lentivirus[10]. HepG2 reporter cells stably transduced with lentiviral CAGA-Luc reporter and Renilla luciferase for signal normalization have been described previously[19]. The HepG2.α1-PDX reporter cell line was derived by transducing HepG2 CAGA-Luc cells with α1-PDX-IRES-GFP lentivirus encoding the PC-inhibitory antitrypsin variant α1-PDX. A clonal cell line was obtained by serial dilution following the sorting of GFP+ cells on a FACSAria Fusion machine (BD Biosciences, San Jose, USA). All cell lines were cultured for maximally 6-8 weeks after de-freezing, and only after testing negative for Mycoplasma infections (Mycospy, Biontex, Munich, DE).

## Site-directed mutagenesis, cDNA cloning, and expression vectors and cloning

Specific mutations in *INHBA* cDNA were generated by amplifying the pLenti-EF1alpha-ActA-SV40-bsd vector using mutagenic primers of interest for overlap extension PCR, together with the external PCR primers 5'-GAGATCTAGACTCGAGACGCAAGGC-3' and 5'-GAGAATC-GATACCGTC GACTAGAG-3' that introduce Xba I and Cla I restriction sites, respectively. PCR products were amplified with external PCR primers and inserted between Xba I and Cla I sites in pLenti-EF1alpha-ActA-SV40-bsd to thereby replace the wild type Activin-A coding sequence.

To stably express α1-PDX in HepG2 cells its coding sequence was inserted into IRESeGFP destination vector (gift of Dr. Joerg Huelsken, EPFL) by Gateway cloning. Briefly, a first PCR was performed to flank α1-PDX with attB1 and attB2 sites. The BP reaction was performed by mixing the resulting PCR product and the pDONR221 Vector (Addgene, Watertown, MA, USA) containing attL1 and attL2 sites using BP Clonase II Enzyme mix (ThermoFisher Scientific). The LR reaction was performed by mixing the obtained Entry Clone with the IRESeGFP Destination vector and with LR Clonase II Enzyme mix (ThermoFisher Scientific). The primers used for mutagenesis and Gateway cloning are listed in Supplemental Table 1.

## Lentiviral transduction

The FurKO#1, #2, #3 and DKO CRISPR clones of B16-F1 cells were transduced with βA or empty control lentivirus as previously described for B16F1-Ctrl and -βA cells[9]. In short, HEK293T cells were co-transfected with CMVΔR8.74 (Addgene, Watertown, MA, USA 22036), pMD2.VSVg (Addgene 12259) mycINHβA, mS1, or empty transfer plasmid. Lentiviral particles were collected from filtered culture supernatant by ultracentrifugation and resuspended in sterile PBS. Target cells were transduced in a 12-well plate. B16-F1 cells were selected using blasticidin. Clonal B16-F1 FurKO- and DKO-βA cell lines were grown from single cells that were plated at limiting dilution, because the CRISPR-edited parental FurKO clones #1 to #3 were already resistant to available antibiotics. Expression of the lentiviral βA transgene was confirmed by Luciferase assay in all the tested clones

($n = 11$). To express βA mutants, lentiviral particles were titrated on B16-F1 and B16F1 FurKO clones #1 and #2 in increasing amount to then select among the resulting cell lines those that express mutant *INHβA* in amounts comparable to wild-type βA in B16F1-βA and B16F1 FurKO-βA cells. B16-F1 IPTG inducible shKlk8 cell lines were generated using pLKO-tGFP-IPTG-1xLacO (Sigma-Aldrich). 5% of the brightest GFP+ cells were pooled after sorting on a FACSAria Fusion machine (BD Biosciences). An shRNA sequence targeting Luciferase was used as negative control. The IPTG-inducible shRNA sequences used in this study are listed in Supplemental Table 2. HepG2.α1-PDX reporter cell line was generated by transducing HepG2 CAGA-Luc cells with viral particles derived from HEK293T cells transfected with CMVΔR8.74, pMD2.VSVg, and α1-PDX-IRES-GFP vectors. Clonal cell line was obtained following sorting of GFP+ cells on a FACSAria Fusion machine (BD Biosciences, San Jose, USA).

### Transfection of plasmid DNA or siRNAs

For DNA transfection, $1 \times 10^5$ cells were seeded in 24-well plates in 500 µL of appropriate medium and transfected 6 hrs post-seeding with 0.5 µg of expression plasmid using Lipofectamine 3000 (ThermoFisher Scientific) following the manufactured protocol. For siRNA transfection, $1 \times 10^5$ or $2 \times 10^5$ cells were plated in 24- or 48-well dishes, respectively, and transfected 1 day after seeding with 10 nM siRNA using INTERFERin (Polyplus, Illkirch-Graffenstaden, FR) following the manufacturer's protocol. Cells and supernatants were collected 72 hrs after siRNA transfection. Negative Control No. 5 siRNA (siNT) (AM4642, ThermoFisher Scientific) and AllStars Mm/Rn Cell Death siRNA (siCyto) (SI04939025, Qiagen) were used as negative and positive controls, respectively.

### Gene expression analysis

Total RNA from cancer cell lines was extracted using ReliaPrep™ RNA Miniprep Systems (Promega), according to the manufacturer's protocol. For bulk tumor RNA extraction, snap-frozen tumor pieces were homogenized in 1 mL of QIAzol (Qiagen, Hilden, DE), followed by chloroform extraction. Total RNA was extracted using RNeasy mini kit (Qiagen), following the manufacturer's protocol. 1 µg of total RNA was reverse transcribed into cDNA using the PrimeScript RT Master Mix (RR036, TaKaRa). RT-qPCR analysis was performed using the PowerUp SYBR Green Master Mix (Applied Biosystems) on a QuantStudio 6 instrument (Applied Biosystems). The PCR and qPCR primers used in this study are listed in Supplemental Table 3. Where indicated, cells were treated for 48 hrs with recombinant Activin-A (R&D Systems, 338-AC) or 10 µM SB 431542 (Sigma-Aldrich, S4317) prior to RNA extraction.

### Western blot analysis

Proteins in cell supernatants were precipitated in ice-cold acetone and resuspended in PBS containing 1 mM EDTA, 0.5% Triton, and protease inhibitor cocktail (Roche). Cells were lysed in RIPA buffer composed of 50 mM Tris pH 8, 150 mM NaCl, 0.1% SDS, 0.5% Na-Deoxycholate, 1% NP-40, supplemented with protease inhibitor cocktail (A32953, Roche) and phosphatase inhibitor cocktail (P0044, Sigma-Aldrich). After sonication with a Sonifier 250 ultrasonic probe (Branson Ultrasonics, Brookfield, USA), protein concentration was determined by BCA assay (ThermoFisher Scientific). Proteins were separated on 9% SDS-PAGE gels under non-reducing conditions and transferred to nitrocellulose membranes using a Trans-Blot Turbo transfer system (BioRad). Membranes were blocked using 5% skim milk (Sigma) in PBS containing 0.1% Tween-20 and stained with primary antibody against Activin-A (1:500, Abcam ab89307), or against Smad2 (1:1000; Cell signaling 3103), pSmad2 (1:1000, Millipore ab3849-I), or γ-tubulin (1:2000, Sigma-Aldrich GTU88). Proteins were revealed with the ChemiDoc imaging system (BioRad) using sheep anti-mouse or anti-rabbit HRP-coupled secondary antibody (GE Healthcare), and Super-ECL reagents

(34076, ThermoFisher Scientific). Activin-A in tumor extracts and plasma was immunoblotted using biotinylated anti-Activin-A antibody (0.5 µg/mL, BAM3381, R&D Systems) and streptavidin-HRP-labeled secondary antibody (016-030-084, Jackson Immunosearch) and ECL reagents (32106, ThermoFisher Scientific). Activin-A processing was quantified using Image Lab 6.1 software (BioRad) to measure the ratios of the indicated non-saturated protein bands. In accordance with Ginefra et al.[15], mature Activin-A was labelled A30 by rounding up (as in the case of A70) the predicted *actual* size ($2 \times 13 + 2\,kDa$ kDa) of the myc-tagged dimer.

### CAGA-Luciferase assay

$3 \times 10^4$ HepG2-CAGA reporter cells were seeded into 96-well plates and treated for 12 hrs with conditioned medium (2 to 10-fold dilution) or, where indicated as positive control, with 20 ng/mL recombinant Activin-A (338-AC, R&D Systems). To block further Activin-A processing in cell SNs during incubation on reporter cells, 100 µM decanoyl-RVKR-chloromethylketone (CMK) (Enzo Life Sciences, Villeurbanne, FR, ALX-260-022) was added where indicated. Cells were lysed in 50 µL 10 mM potassium buffer containing 0.2% Triton-X100, and the luminescence of Firefly and Renilla were each measured using a Centro XS3 LB 960 luminometer (Berthold Technologies) by adding 5 µL of each lysate to a mixture of 50 µL P/R A Firefly reagent (25 mM DTT (Huberlab, A11010005), 1 mM ATP (Sigma, A2383), 0.2 mM coenzyme A (Sigma, C3144), 200 µM Luciferin (Biosynth, FL08608) in Luciferase buffer pH 8.0 (15 mM MgSO$_4$, 0.1 mM EDTA, 200 mM Tris)) and 50 µL P/R B Renilla reagent (50 µM APMBT (Calbiochem, 444350) and 4 µM benzyl-coelenterazine (Focus Biomolecules, FMB-10-1496)) in Renilla buffer pH 5.0 (10 mM NaOAc, 15 mM EDTA, 500 mM Na$_2$SO$_4$, 25 mM Na$_4$PPi)) in white Nunc 96-well plates (ThermoFisher Scientific, 236108). Firefly luciferase measurements were normalized using Renilla values. Relative light units (RLU) represent normalized expression relative to non-treated control.

### siRNA library screen

After identifying optimal cell numbers and siRNA concentrations for the siRNA screen, B16F1 FurKO#1-βA cells were transfected with the G-015105 Mouse protease siRNA library (Horizon Discovery, Waterbeach, UK) at a density of $0.5 \times 10^3$ cells per well in 96-well plates using 0.5 µL of INTERFERin (Polyplus) per well, according to the manufacturer's protocol. Cells were transfected using 10 nM of a pool of four siRNA sequences per gene in duplicate. The siRNA sequences from the library G-015105 used to validate candidates are listed in Supplemental Table 4. Murine siPcsk5 and siPcsk6 were purchased from Horizon Discovery (Supplemental Table 5). Negative Control No. 5 siRNA (siNT) (AM4642, ThermoFisher Scientific) and AllStars Mm/Rn Cell Death siRNA (siCyto) (SI04939025, Qiagen) were used as negative and positive controls, respectively. Three days after siRNA transfection, the effect on furin-independent Activin-A signaling was quantified by adding to each well $1.5 \times 10^4$ HepG2.α1-PDX reporter cells that express SMAD3-inducible Firefly luciferase (CAGA-Luc) together with Renilla luciferase for signal normalization. To estimate the signal activation by the melanoma cells themselves, coculture SNs were set aside after 24 hrs, followed by cell lysis to measure the induction of Firefly luciferase relative to Renilla as described above. For signal normalization, we also estimated the total activity of Activin-A (i.e. including the signaling unleashed by Activin-A cleavage during co-culture with HepG2 reporter cells) by transferring one tenth (20 µL) of the co-culture SNs to 96-well plates containing HepG2 CAGA-Luc reporter cells without α1-PDX ($3 \times 10^4$ cells/well). After incubation for 12 hrs, the reporter cells were lysed to measure the expression of Firefly and Renilla luciferase. The signaling activity of Activin-A that was pre-cleaved by PCLP in melanoma cells was then normalized to the total activity, by calculating the ratio of the fold change of CAGA-Luc induction in HepG1.α1-PDX relative to the fold change in control HepG2 reporter cells. The Z'

factor was calculated with mean and SD of the positive and negative controls using the formula $(1-(3*SD_{siNT} + 3*SD_{siCyto}))/(mean_{siCyto}-mean_{siNT})$[66].

## Colony formation assay in soft agar

Low melting agarose (Sigma-Aldrich, A9414) at a concentration of 1% was mixed with an equal volume of medium containing 2X additives. The resulting 0.5% agarose-1X medium mix was poured in 6-well plates (1.5 mL/well) and stored at 4 °C. The next day, $2 \times 10^3$ cells were seeded in 0.35% agarose-1X medium mix. Plates were incubated for up to 14 days at 37 °C and then stained for 1 hr with crystal violet (Sigma-Aldrich). Excess crystal violet was removed and wells were washed once with PBS. Pictures were taken using a camera mounted on a light table. Macrocolonies were counted manually using the QuPath software.

## Analysis of cell viability by AlamarBlue staining

$2 \times 10^3$ cells were seeded into 96-well plates in 200 μL complete medium. After proliferation during 3 days, the number of viable cells was quantified by adding 20 μL AlamarBlue reagent per well (Invitrogen, Waltham, USA). After incubation at 37 °C for another 3 to 4 hrs, AlamarBlue fluorescence at 560 nm excitation was measured at the emission wavelength of 590 nm using an Infinite F500 microplate reader (Tecan, Männedorf, CH).

## Analysis of cell death by Annexin V/PI staining

To analyze the cell death induced by the siKlk8 transfection, $3 \times 10^4$ B16F1 FurKO#1-βA melanoma cells were seeded in 24-well plates. 1 day after seeding, cells were transfected with siNT or siKlk8 and stained after another 3 days with Annexin V apoptosis detection kit (BioLegend, San Diego, USA, 640932), according to the manufacturer's protocol. In short, cells were washed and resuspended in Annexin V binding buffer, and stained with Annexin V FITC and propidium iodide (PI) for 15 min at room temperature in the dark. After adding Annexin V buffer to inactivate Annexin V, cells were immediately analyzed using LSR Fortessa cytometer (Becton Dickinson, Franklin Lakes, USA).

## In vitro cleavage of proActivin-A by recombinant proteases

100 μL of FurKO#3-βA cell SN containing WT or mutant A110 was acidified with 1 M HCl to a final concentration of 70 mM (final sample pH 2). After 1 hr at room temperature, Hepes buffer (Gibco 15630-080) and NaOH were added at final concentrations of 25 mM and 70 mM, respectively, to reset the pH to 7. Non-acidified samples were only supplemented with Hepes buffer alone. rKLK6 (R&D Systems, 5164-SE) and rKLK8 (R&D Systems, 2025-SE) were activated by lysyl-endopeptidase (SERVA Electrophoresis, Heidelberg, DE) for 30 min at 37 °C. rKLK12 (R&D Systems, 3095-SE) was activated for 24 hrs at 37 °C in 100 mM Tris-HCl pH 8 containing 150 mM NaCl, 10 mM $CaCl_2$, and 0.05% Brij35 (ACR32958, ThermoFisher Scientific). rKLK5 (R&D Systems, 1108-SE) did not need any activation step. 25 mM of the indicated in vitro-activated recombinant kallikreins, or 4 U of recombinant human furin (rFurin, New England Biolabs) lacking the transmembrane domain and cytosolic tail were incubated with the cell SN for 5 hrs at 37 °C, followed by Western blot analysis of Activin-A processing.

## Cleavage of fluorogenic AMC substrates

0.5 ng of in vitro-activated rKLK8 was incubated in 100 μL KLK8 Assay buffer (50 mM Tris-HCl, pH 9) for 1 hr at 37 °C, together with 0.1 mM fluorogenic Boc-VPR-AMC substrate (R&D Systems, ES011). 0.5 ng of rKLK5 was incubated in 100 μL KLK5 Assay buffer (0.1 M $NaH_2PO_4$, pH 8) for 1 hr at 37 °C, together with 0.1 mM fluorogenic Boc-VPR-AMC substrate (R&D Systems, ES011). 0.5 ng of in vitro-activated rKLK6 was incubated in 100 μL KLK6 Assay buffer (50 mM Tris-HCl containing 1 M sodium citrate, pH 7.5) for 1 hr at 37 °C, together with 0.1 mM fluorogenic Boc-QAR-AMC substrate (R&D Systems, ES014). 0.5 ng of rKLK12 was incubated in 100 μL KLK12 Assay buffer (100 mM Tris-HCl pH 7.5 containing 150 mM NaCl, 10 mM $CaCl_2$, and 0.05% Brij35) for 1 hr at 37 °C, together with 0.1 mM fluorogenic Boc-VPR-AMC substrate (R&D Systems, ES011). 2 U of rFurin was incubated in 100 μL Furin Assay buffer (0.5% Triton X-100, 1 mM $CaCl_2$, 1 mM β-mercaptoethanol in 100 mM Hepes, pH 7.5) for 1 hr at 32 °C together with 0.1 mM fluorogenic Boc-RVRR-AMC substrate (Enzo Life Sciences, ALX-260-040). Fluorescence of Boc-VPR-AMC or Boc-QAR-AMC was measured at 380 nm (excitation) and 460 nm (emission), or at 355 nm (excitation) and 440 nm (emission) for Boc-RVRR-AMC, using a Safire II microplate reader (Tecan). Where indicated, CMK or D6R (344931, Merck) were added at the indicated concentrations, and the fluorescence signals were normalized to the values of rKLK8 or rFurin without inhibitor.

## LC-MS/MS mass spectrometry

The peptide SHRPFLMLQARQSEDHPHRRRRRGLEQDGKVNICCKKQFFV comprising the furin cleavage motif RRRRR and 18 flanking residues on either side from proActivin-A, and an analogous peptide containing the mutant RRRRA motif were synthesized by the peptide & tetramer core facility of the University of Lausanne. Peptides were incubated for 5 hr at 37 °C at a concentration of 0.4 μg/μL in 10 μL Tris-HCl pH 7.5 containing 1 mM $CaCl_2$ and 16 nM rKLK8, or 2 U rFurin, or 0.1 μg/mL lysyl-endopeptidase. After incubation, peptides were desalted on C18 StageTips[67] and dried by vacuum centrifugation prior to LC-MS/MS injections. Samples were resuspended in 2% acetonitrile (Biosolve), 0.1% FA and nano-flow separations were performed on a Dionex Ultimate 3000 RSLC nano UPLC system (ThermoFisher Scientific) on-line connected with an QExactive Orbitrap Mass Spectrometer (ThermoFisher Scientific). A capillary precolumn (Acclaim Pepmap C18, 3 μm-100Å, 2 cm x 75μm ID) was used for sample trapping and cleaning. A 50 cm long capillary column (75 μm ID; in-house packed using ReproSil-Pur C18-AQ 1.9 μm silica beads; Dr. Maisch) was then used for analytical separations at 250 nl/min over 90 min biphasic gradients. The mobile phases were as follows: A, 2% acetonitrile, 0.1% formic acid in water and B, 90:10 (v/v) acetonitrile:water, 0.1% formic acid. The mass spectrometer was operated in positive data-dependent acquisition mode, and the full MS range was from 300 to 2000 m/z. The 7 most intense ions were isolated in the quadrupole and fragmented under high-energy collisional dissociation with a normalized collision energy of 27% with a 30 s exclusion list. Precursor and fragment ions were measured at a resolution of 70,000 and 17,500 (at 200 m/z) respectively. Only ions with charge states of 2 and higher were fragmented with an isolation window of 1.2 m/z. Peptide fragments were identified by the PEAKS search engine using the software PEAKS Studio 10.6 (build 20201221) with the Parent and Fragment Mass Error Tolerances set at 15.0 ppm and 0.05 Da, respectively. Precursor Mass Search Type was mono-isotopic, with the Digest Mode set as "Unspecific", and with maximally two modifications by oxidation (M 15.99) per peptide. FDR Estimation: Enabled. Merge Options: no merge. Precursor Options: no correction. Charge Options: no correction. Filter Options: no filter. Process: true. Associate chimera: no. A complete list of the identified peptides can be found in the Supplementary Data 1.

## Analysis of melanoma grafts in syngeneic mice

The indicated B16-F1 melanoma cell lines were injected intradermally at a density of $1 \times 10^5$ cells in 25 μL phosphate-buffered saline into the right flank of 9 to 10-week-old female C57BL/6J mice (Charles River laboratories, Wilmington, USA) that were housed at EPFL in a 12-h-light-12-h-dark cycle, controlled temperature (22 ± 2 °C) and humidity (55 ± 10%), and with food and water ad libitum. To deplete Klk8 in tumors lacking furin, we chose the B16F1 FurKO#2 clone after confirming that its tumor growth rate was comparable to that of clone #3 and of parental B16-F1 cells[16]. For *Klk8* depletion, 20 mM IPTG

(AppliChem) was supplied in drinking water together with 50 g/L sucrose (AppliChem), starting 1 day after tumor grafting[68]. Drinking water was replaced in 48 hrs intervals. Tumors were measured every other day until the endpoint to calculate tumor volumes using the conservative estimate V = [1.58π x (length x width)$^{3/2}$]/6[69]. The maximal tumor size was 1 cm³, except in rare cases that were authorized individually by the cantonal veterinarian based on the well-being of the animals and absence of tissue ulceration and necrosis. For flow cytometric analysis, tumors were dissected, minced using rounded scissors, and digested in Dnase-I (0.02 mg/mL, Sigma) and collagenase (1 mg/mL, Sigma) in RPMI using a gentleMACS Octo Dissociator (Miltenyi). Red blood cells were lysed using PharmLyse buffer (BD Biosciences). After washing in PBS, 1-5 × 10⁶ of cells were used for staining. Cells were incubated with mouse FcR blocking solution (1:200, Miltenyi) and with Live/Dead fixable blue dead cell stain (1:1000, Life Technologies, Carlsbad, CA, USA) for 30 min. After washing, cell surface markers were labeled with specific antibodies for 45 min in FACS buffer (2% FBS, 2 mM EDTA in PBS). After the staining, cells were collected in FACS buffer, and data were acquired using a LSR Fortessa cytometer (Becton Dickinson, Franklin Lakes, NJ, USA). The antibodies used for the stainings are listed in Supplemental Table 6. Data were analyzed using FlowJo (v.10.8.1, Becton Dickinson). The gating strategy used to quantify the CD8⁺ T cells is shown in Fig. S5F.

## Quantification of Activin-A plasma levels

Blood was collected in heparin-coated tubes (Microvette 500 LH, SARSTEDT AG & Co.) at the experimental endpoint. Plasma was then separated by centrifugation and aliquoted into new Eppendorf tubes. Activin-A levels in plasma were quantified by ELISA following the manufacturer's instructions (R&D Systems, DAC00B).

## Whole mount immunofluorescent staining

Tumors were embedded in OCT (ThermoFisher Scientific). 10 μm cryosections were completely dried, rehydrated for 10 min in 1X PBS, and fixed in 4% PFA (Electron Microscopy Sciences, Hatfield, USA, 50-980-494) for 10 min at RT. After washing, sections were permeabilized with 0.25% Triton X-100 (AppliChem) for 10 min, washed again, and incubated for 1 hr at RT in 1% BSA (Sigma-Aldrich) as a blocking agent, followed by overnight incubation at 4 °C with antibodies against Ki-67 (ThermoFisher Scientific, MA5-14520, 1:100) or cleaved caspase 3 (Cell signaling, 9661, 1:100). After washing away unbound antibody, sections were incubated with fluorescent secondary α-rabbit-Alexa568 antibody (ThermoFisher Scientific, A10042, 1:800) and counterstained with 0.1 μg/mL DAPI. Sections were mounted on slides with Fluoromount Aqueous Mounting Medium (Sigma-Aldrich, F4680). Images were acquired with a LSM710 confocal microscope (Zeiss), analyzed using the ImageJ software (Fiji), and quantified using QuPath.

## RNAscope analysis of Klk8 mRNA expression

B16F1-βA shLuc, shKlk8_1, and shKlk8_4 cells were treated for 4 days with 100 μM IPTG. 1.25 × 10⁵ cells were then seeded into chambers of Culture Slides (Falcon, 354114) and cultured for 2 additional days in 100 μM IPTG-containing medium. RNAscope Multiplex Fluorescent V2 assay (Biotechne, 323110) was performed according to manufacturer's protocol (Advanced Cell Diagnostics, Newark, USA). Briefly, cells were fixed with 10% neutral buffered formalin (Sigma-Aldrich, HT501320), and hybridized with the probes Mm Ppib-C1 as positive control (Biotechne, 313911), Dapb-C1 (Biotechne, 310043), or Mm Klk8 probe (Biotechne, 515251) at 40 °C for 2 hrs. The probes were revealed with TSA Opal570 (Akoya Biosciences, FP1488001KT). Cells were counterstained with DAPI and mounted with Prolong Gold Antifade Mounting medium (ThermoFisher, P36930). Images were acquired with a widefield microscope (Zeiss), and analyzed using the ImageJ software (Fiji).

## Bioinformatics analysis

TCGA Melanoma (SKCM) gene expression RNAseq (HiSeqV2), clinical and survival data were downloaded from the UCSC Xena Hub using the *UCSCXenaTools* R package (v. 1.4.7)[70]. The selected patients included those with primary tumor or metastatic disease and with survival data, but with no history of neoadjuvant treatment. Additional information about the group sizes and boxplot values can be found in the Supplementary Data 2. The association between *INHβA* expression or disease stages and *KLK8* expression groups was computed. Kaplan-Meier plots were generated by splitting patients in two groups based on *KLK8* expression (*KLK8* expression = 0, *KLK8* expression > 0). For the twelve other cancer types, Kaplan-Meier plots of patients stratified by *KLK8* expression levels were obtained using KM-plotter (TCGA)[71]. Tissue distribution of *KLK8* expression and *KLK8* mRNA expression data in human cancers were obtained from the Human Protein Atlas database (https://www.proteinatlas.org/). *KLK8* mRNA levels in tumor types and in corresponding healthy tissues were obtained from the TCGA database using the FireBrowse server (https://firebrowse.org/).

## Statistical analyses

All statistical test were performed using the Prism software (GraphPad). Unless indicated, data represent mean ± SD of at least 3 independent experiments. When comparing two groups, normal distributions were analyzed by the Shapiro-Wilk normality test, and *p*-values calculated by Student's *t*-test (normal distribution) or Mann-Whitney's test (non-parametric test). One-way ANOVA was used to compare several groups of unpaired values. Kaplan-Meier survival curves were analyzed using the Gehan-Breslow-Wilcoxon test. Data points identified as outliers by the regression and outlier (ROUT) removal method in Prism 9 with a False Discovery Rate ≤1% were excluded. Power Analysis was waved by the animal experimentation authorities due to pre-existing data about the effect sizes of Activin-A induced tumor growth and loss of body weight. Tumor growth curves were compared by Two-way ANOVA.

## Reporting summary

Further information on research design is available in the Nature Portfolio Reporting Summary linked to this article.

# Data availability

The mass spectrometry proteomics data have been deposited to the ProteomeXchange Consortium via the PRIDE partner repository with the dataset identifier PXD060918. The remaining data are available within the Article, Supplementary Information. Request for resources and reagents should be sent to and will be answered by the lead contact Prof. Daniel B. Constam (daniel.constam@epfl.ch). Source data are provided with this paper.

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

## Acknowledgements
The authors are grateful to Drs. Jonathan Pittet and Maria Pavlou of the EPFL Proteomics core facility for their support with the analysis and identification of in vitro-cleaved peptides. We also would like to thank Dr. Jessica Dessimoz of the EPFL Histology Core Facility for advice with RNAscope analysis, and Gian-Filippo Mancini for tumor sectioning and staining with the RNAscope probes. We also thank the members of the Center of PhenoGenomics for the animal housing and help with IPTG renewal in the drinking water, Drs. Anna C. Obenauf (Vienna, Austria) for providing YUMM3.3 cells, Dr. Joerg Huelsken for providing the IRES-eGFP destination vector, and the Constam lab members for comments on the manuscript. This work was supported by grants 31003A_179330 of the Swiss National Science foundation and KFS-4454-02-2018 from the Swiss Cancer League to DBC, and by services of the Center of PhenoGenomics and Flow Cytometry Research Core Facilities at the School of Life Sciences of EPFL.

## Author contributions
Conceptualization: M.B., K.P., and D.B.C.; methodology: M.B. and D.B.C.; bioinformatics analysis: N.F.; validation: M.B.; formal analysis: M.B.; investigation: M.B., L.I., C.S.; writing—original draft: M.B. and D.B.C.; review and editing: M.B., K.P., D.B.C.; visualization: M.B., N.F.; supervision, project administration, funding acquisition, and guarantor: D.B.C.

## Competing interests
The authors declare no competing interests.
