## [Transparent Peer Review file · Nature Communications]

Kallikrein-8 mediates furin-independent Activin-A precursor processing to stimulate tumor growth in melanoma

Corresponding Author: Professor Daniel Constam

Version 0:

Reviewer comments:

Reviewer #1

(Remarks to the Author)

The study highlights the independent function of KLK8 in cleaving proactivin-A to A70 and underscores the critical role of the R310 site and its surrounding structural environment. This contributes significantly to a deeper understanding of the cleavage mechanisms of activin-A, however, further enhancement in the rigor of methodology, reliability of data, and depth of result interpretation is necessary to meet the high publication standards of Nature Communications. Below are my suggestions for revision:

1. The manuscript would benefit from a more detailed discussion of the role of Activin-A in diseases, particularly in cancer and fibrosis in the introduction section, to better contextualize the biological and clinical significance of the study.
2. The manuscript lacks sufficient biological validation regarding the specific mechanisms by which Klk8-mediated cleavage of activin-A affects tumor growth.
3. The manuscript convincingly shows that furin plays a significant role in cleaving proactivin-A to generate A30 and A60. However, the conclusion in Fig. 6G that furin can also cleave A70 to A60 lacks direct evidence. This claim should either be supported by additional experimental data or suitably moderated in the text.
4. In Fig. 1E, the rationale for selecting specific siRNAs to validate the A70/A110 ratio is not clear.
5. The YUMM3.3- β A group shows considerable variability within the dataset in Fig. 3F. It is recommended to investigate the cause of this variability and ensure experimental consistency.
6. Attention should be paid to the labeling of protein bands for A110, A70 and A30, especially in figures such as Fig. S1B and S1F, where the position marked for A30 appears to be below 25kDa, which could confuse readers.
7. It is advisable to use asterisks to denote levels of statistical significance between groups to enhance clarity and readability.

Reviewer #2

(Remarks to the Author)

The manuscript by Buillard et al describes a novel mechanism for proteolytic activation of activin A (usually spelled without a hyphen) from its pro-forms by kallikrein-8. Activin A, like other TGF- β family proteins, is produced in precursor form which is processed at least by one proteolytic enzyme, typically furin or furin-like proprotein convertase, in a polybasic site separating the pro- and mature domains. Authors have previously shown that knocking out furin results in substantial suppression of proteolytic processing of pro-activin A. However, some processing does occur, to a partially (hemi) cleaved form, but the identity of the protease responsible for this was not known. This current manuscript seeks to address that.

The authors performed an siRNA screen of known protease genes which suggested that the likely candidate responsible for pro-activin A processing is kallikrein-8 and most of the manuscript deals with verification of this finding.

While identification of the alternative processing of pro-activin A is interesting, the manuscript fails fully to address the biological relevance of this processing, or to propose a mechanism to explain why kallikrein-8 would cleave just one of the two linkers between the pro- and mature domains in the dimeric growth factor. We are not sure how much of this finding is relevant in biology and not simply observed in experimental systems in which activin A is over-expressed.

Some of our scepticism arises from the authors' own interpretation of their data. For example, in the siRNA experiments even control cells with non-targeting siRNA (siNT) fail to process activin A correctly and this treatment reduced activin activity by 50%. If the latter was really due to "edge effects" as authors suggest, did they run these samples in the middle of

the plate to confirm it? Indeed, if there was a significant edge effect (which could be mitigated), we hope the whole experiment was repeated with samples in different places. Little information was provided about cell viability as well in these early experiments, which was an issue later. What was the effect of KLK8 siRNA treatment on cell survival. Perhaps authors can provide data for that.

KLK8 was also not the only gene (or pool for that matter) that had significant effect on pro-activin A processing. What happened to Adamts10? Its inhibition had a similar effect to KLK8, but it seems to have been dropped off as a potential candidate without validation. Will there be a separate publication on that?

The authors also conducted a validation experiment in which cell culture supernatants containing activin A were treated with recombinant kallikrein-8. Processing by kallikrein-8 required acidification of the sample to a pH of at least 6.5, but data were shown for pH 2. At this low pH, we struggle to see how this processing could be physiologically relevant or even that it is not an artefact resulting from pH-driven changes in protein structure. Acidification is routinely used to release TGF- β from its latent form, but reduction in processing by furin could be simply an indication that treatment is causing activin A to behave non-natively and not a demonstration of any specific mechanism.

One of the key experiments underlying the story is an siRNA screen against 542 proteases. This was performed in a cell line overexpressing activin A, using lentiviral transgene. It is not clear what level the expression of activin A is in these cells, how it relates to physiological levels and how overexpression may affect activin A processing. Some of the interpretations are also partly dependent on a parallel manuscript by the same authors on alternative forms of activin A. The authors discuss the non-canonical 58 kDa (A60, see later comment on nomenclature) cleavage product, but the relevance of this is not clear. It is well known that overexpression of these proteins in mammalian cells can result in high molecular weight aggregates and secretion of partially processed forms. The latter can be corrected by furin overexpression, suggesting that the intrinsic capacity of cellular machinery is limited for proprotein processing. Of note in this regard are the higher molecular weight bands which are marked as "ns" (non-specific) in some of the western blots. It would be appropriate to confirm that they are truly result of non-specific antibody reactivity and NOT activins in aggregated state.

The authors then use mutagenesis of the polybasic furin site (overlapping with putative kallikrein recognition/cleavage site) to interrogate the key residues required for cleavage by kallikrein-8. We are not convinced that the mutagenesis experiments conducted provide conclusive evidence that the R310 specifically favours cleavage by kallikrein-8 as described.

The authors first show that mutation of R310 to alanine abolishes kallikrein-8 processing of pro-activin A. They then speculate this is possibly due to changed sequence context of the processing site and examine this hypothesis by moving the rest of the sequence up by one, resulting in sequence RRRRALE (R310A mutation, elimination of _native_ Gly from RRRRRGLE). Why didn't the authors just remove the 5th arginine, rather than retain the mutated alanine and remove the natural glycine? This change is not very large, agreed, but at the same time it does not fully replicate the sequence context of the native cleavage site. The resulting discussions on structural context of the site are simply nonsensical – with the exception of MSTN (where the entire furin site is seen in the crystal structure), the furin site is poorly structured (flexible) and not resolved in other TGF- β structures.

When it comes to biological validation of the possible kallikrein-8 processing of pro-activin A, the work relies again on overexpression of activin A to see the effect – cells without overexpression do not show difference in tumour growth. Is this physiologically significant?

In these experiments (Fig 5 and 6) the western blots in particular could do with better labelling as sample numbers are not identical in all cases for tumour and serum samples. Also, why is the western blot in Figure 6 suddenly showing mostly proteins at much higher molecular weight, well beyond 110 kDa species of activin A, compared to other experiments?

Finally, we also wanted to address authors' use of abbreviations, especially for the different forms of activin A. They observe uncleaved 110 kDa form, hemi-cleaved 66 kDa form, alternative hemi-cleaved form at 58 kDa and the mature growth factor of 26 kDa (which by the way runs below 25 kDa on all of the gels!) in non-reduced SDS-PAGE/western blot analyses. Why are the names for the different forms different from the observed molecular weights? Why is 66 kDa form A70, 58 kDa A60 and mature domain as A30 (why not just call mature activin A as that, like is customary). This adds unnecessary level of convoluted and certainly follows no expected standard. It simply makes no sense to round up these numbers. The manuscript is also otherwise a riot of unique abbreviations : SN for supernatant, DKO for furin/PC7 double knock-out, FurKO for furin knock-out, PCLP for PC-like protease (google search comes up with "podocalyxin-like protein")... There is a time and place for abbreviations and saving space, but it cannot be at the expense of readability and accuracy. There is also some inconsistency in using gene and protein names in the text.

Overall, we are on the fence about this manuscript. There is a significant amount of work in this manuscript which is mostly of high quality and kallikrein-8 cleavage has been shown convincingly in this experimental setup. However, we are not convinced about the significance of these findings either in normal biological context or in disease.

Reviewer #3

(Remarks to the Author)

Reviewer #4

(Remarks to the Author)

I find this a comprehensive study showing that KLK8 participates in the stepwise cleavage of Activin-A precursor protein which seems to be involved in the progression of melanoma tumours.

I have two comments that I think may improve the quality of the manuscript:

1. Figure 4C: The authors claim they show representative figures of crystal violet staining of cells at the experimental endpoint. As I understand the staple diagrams, there are stronger signals in the lower panel. When I look at the photographs included, I can see more spots indicating stained cells in the upper panel, and no spots at all in the two pictures to the right in the lower panel.

2. Discussion section "KLK8 can convert proActivin-A to hemicleaved form": I agree with the authors concerning the overall conclusions drawn in the discussion, but I have some issues concerning how some references are used in the context as exemplified below:

- Line 411 ref (34) is misleading in this content. In this paper, the inhibition of KLK5, KLK7 and KLK14 by LEKTI fragments are studied, and it is shown that this inhibition is pH-dependent which is referred to in the text. But KLK8 is not included in that study. In Eissa et al (ref 32) it is mentioned that KLK8 is not inhibited by any LEKTI fragments.
- Line 417-418 it is stated "...a pH-regulated proteolytic cascade initiated by KLK5 is important for the zymogenic maturation of KLK8" and the ref for this is Eissa et al (32). I agree with the authors that this may be the case, but in the paper of Eissa et al they show that KLK8 is active also during acidic conditions, but all activation experiments were performed at basic conditions. The idea of a pH regulated proteolytic cascade initiated by KLK5 was initially published in Brattsand et al, *J Invest Dermatol* 124: 198-2003, 2005, where it was shown that the activation rate of proKLK7 by KLK5 peaked at pH 5.5 although the pH optimum for the KLK5 activity towards chromogenic substrates were slightly basic as for KLK8. Therefore, it could be hypothesized that the activation of proKLK8 by KLK5 could be affected by pH although it has not yet been published to my knowledge.

Version 1:

Reviewer comments:

Reviewer #1

(Remarks to the Author)

Authors have fully addressed my concerns. I suggest it for publication.

Reviewer #2

(Remarks to the Author)

We would firstly like to give credit to authors for comprehensive replies to our (and other referees') comments. We also applaud them for re-running experiments to ensure consistency of data.

We are still in two minds with the manuscript. There is no arguing with the technical quality and amount of work in this manuscript. The authors have left hardly any stone unturned to confirm their findings. However, most of the queries which relate to the biological/medical significance (e.g. relation to human cancers) of this specific work on KLK8's role in activin signalling are still speculative to us. One of the reasons for this relates to our original comment on the results of the protease siRNA screen where number of other proteases were hits, in addition to KLK8 (incl. ADAMTS10 that was discussed in rebuttal letter). As several other proteases are able to process activin A as well, is this a case of more general (promiscuous, serendipitous?) processing of pro-activin A by extracellular proteases rather than truly KLK8 specific one? Data from the negative control siRNA (siNT) shows on activin, again suggesting more unspecific effect at play here. We feel these doubts on biological significance (shared partly by the other reviewer) lower the impact and importance of this work.

One reason for this has taken a while for us to review a little longer (now and earlier) is to do with somewhat convoluted way the manuscript and rebuttal letter are written. This is partly to do, as we raised before, with the fact that manuscript is littered with non-standard abbreviations (while non-standard activin-A with hyphen is justified as one reducing confusion). If someone earlier has come up with non-sensical naming of proteins based on the rounded-up molecular weights, we don't see a need to follow that. However, this is of course authors' choice and we need not argue with that. Our comments in this regard were simply trying to help to make this manuscript more accessible to read.

As we have not issues with technical aspects of the manuscript, we are happy to recommend this to be published in the revised form, but we are not sure if it quite at the level of significance to warrant Nature Communications.

The only revision we would recommend authors to make is to do with the statement on apparent molecular weight of activins changing depending on acrylamide concentration in SDS-PAGE (page 18, lines 627-628). Migration of proteins change on different percentage acrylamide, but proteins keep running in relation to their MW (or rather, hydrodynamic radius). There are protein-specific effect, e.g. with overall charge, but the apparent MW does not change from gel to gel. Otherwise MW markers would need to be different for different percentage gels. So, unless the authors have data showing convincingly that activin A changes its apparent MW (rather than migration) in different acrylamide concentrations or that the whole premise of SDS-PAGE since Laemmli is proven wrong, we would recommend to re-phrase or even remove the statement: "apparent molecular weight that varies depending on the acrylamide concentration" (lines 627-628).

Reviewer #3

(Remarks to the Author)

I co-reviewed this manuscript with one of the reviewers who provided the listed reports. This is part of the Nature

Communications initiative to facilitate training in peer review and to provide appropriate recognition for Early Career Researchers who co-review manuscripts.

Reviewer #4

(Remarks to the Author)

I am satisfied with how the authors have responded to my concerns and have no further questions

Version 2:

Reviewer comments:

Reviewer #2

(Remarks to the Author)

Not sure why we need to spend more time on this manuscript. Like said before, the work is technically comprehensive and high quality. Surely this stage is for the editors to make the decision.

With respect to this latest revision, the authors have, in response to our comments, included several clarifications and limitation to their discussions, which are appreciated.

We are fine for this to be published, let the time decided the significance and relevance.

Reviewer #3

(Remarks to the Author)

KLK8 MS revisions

As you will see from the reports copied below, the reviewers raise important concerns. We find that these concerns limit the strength of the study, and therefore we ask you to address them with additional work. Without substantial revisions, we will be unlikely to send the paper back to review. In particular, we would expect your revised manuscript to address the physiological relevance of KLK8-mediated processing of activin A (Reviewers #2-3), the underlying mechanism of hemi-cleavage (Reviewers #2-3) and its relevance to tumor growth (Reviewer #1), as well as the edge effect in the siRNA experiments and the sequence context of the cleavage site (Reviewers #2-3).

Please provide uncropped scans of all blots (showing membrane edges and protein markers run together) for the reviewers.

Uncropped scans of all blots have been provided as source data.

This is not to say, however, that we consider any other points raised by our reviewers less important, and would expect all of their concerns to be addressed.

Reviewer #1 (Remarks to the Author):

The study highlights the independent function of KLK8 in cleaving proactivin-A to A70 and underscores the critical role of the R310 site and its surrounding structural environment. This contributes significantly to a deeper understanding of the cleavage mechanisms of activin-A, however, further enhancement in the rigor of methodology, reliability of data, and depth of result interpretation is necessary to meet the high publication standards of Nature Communications. Below are my suggestions for revision:

1. The manuscript would benefit from a more detailed discussion of the role of Activin-A in diseases, particularly in cancer and fibrosis in the introduction section, to better contextualize the biological and clinical significance of the study.

In the first paragraph of the Introduction, we now mention the role of Activin-A in fibrosis and in squamous cell carcinoma of the skin, and we cite a recent review by (Cangkrama et al., 2020). Since that excellent review focuses on these subjects in skin and other organs much more ably than we could do here, we think citing it is more appropriate than trying to cover this topic in our study of a melanoma model where fibroblasts are scarce and have not been implicated, and which we think is not related to fibrosis.

2. The manuscript lacks sufficient biological validation regarding the specific mechanisms by which Klk8-mediated cleavage of activin-A affects tumor growth.

The thrust of this manuscript is the identification of KLK8 as a novel regulator of Activin-A signaling, which means that as a mechanism, we had to focus on the role of this protease in Activin-A precursor processing. In line with our *in vitro* data that KLK8 mediates furin-independent hemicleavage, we here showed that depletion of KLK8 significantly enriched uncleaved precursor (A110) relative to mature form (A30) also in tumors (**Fig. 5F, G**). A similar trend was also observed for the A70/A30 ratio, consistent with the prediction of our model that at least one subunit of Activin-A is cleaved independently of KLK8 by furin.

The function of Activin-A itself depends on the context, including the state of its diverse target cells and their relative frequencies. Our current understanding of the pleiotropic functions of Activin-A and their relative contributions to cancer progression is still incomplete and elucidating them further is an arduous task that is pursued in separate projects both by our

lab and many others and hence beyond the scope of this manuscript. This being said, we here confirmed that:

- i. the protective effect of *Klk8* knockdown was strictly dependent on the presence of Activin-A (**Fig. 5B**);
- ii. *Klk8* depletion enriched tumors for CD8⁺ T cells (**Fig. S5B**);
- iii. this increase in CD8⁺ T cells correlated with an increase in the frequency of apoptotic cells marked by cCasp3 staining (**Fig. S5C**), further corroborating that the primary function of KLK8 is to activate Activin-A;
- iv. the rate of cancer cell proliferation marked by Ki67 staining remained unchanged (**Fig. S5D**). This is consistent with the proposed mechanism as well, since the stimulation of tumor growth by Activin-A is mediated by depletion of cytotoxic T cells, and not associated with increased proliferation (Donovan et al., 2017; Pinjusic et al., 2022).

Nevertheless, prompted by the suggestion of this reviewer to further address the mechanism, we now also examined the effect of *Klk8* RNAi on Smad2 activation downstream of Activin-A. In the **new figure S5E** of the revised manuscript, we now show that *Klk8* depletion also significantly decreased the levels of pSmad2 relative to total Smad2. These data further corroborate the proposed mechanism that *Klk8* acts directly on Activin-A.

3. The manuscript convincingly shows that furin plays a significant role in cleaving proactivin-A to generate A30 and A60. However, the conclusion in Fig. 6G that furin can also cleave A70 to A60 lacks direct evidence. This claim should either be supported by additional experimental data or suitably moderated in the text.

That furin prevents A70 accumulation by converting it to A60 can be seen most clearly in cell-free cleavage assays (**Fig. S2A**) and in cells depleted of endogenous furin (**Figs. 2A and S1B**). Although not shown in this manuscript, we also see that A70 is further processed when added to HepG2 reporter cells, unless the latter are treated with furin inhibitor (CMK) or transduced with the proprotein convertase (PC)-specific antitrypsin variant α 1-PDX (Pinjusic et al., *Comms. Biol.*, under revision, Fig. 2C, panel (i), third blot from left).

The reviewer did not spell out an alternative testable model how the loss of furin could abrogate A60 formation and induce A70 other than by blocking at least the cell-autonomous conversion of A70 to A60 (as described previously and as shown here again, e.g. in the *siPcsk3* lane in Fig. S1B). Nevertheless, we did initially consider an alternative scenario whereby A70 could be a product of *aberrant* disulfide bond formation due to inhibition of furin cleavage. However, this is highly unlikely since disulfide bonds form in the ER before Activin-A encounters active furin in the TGN. Furthermore, in the co-submitted manuscript, we found that:

- i. Even in cell-free assays, treatment with recombinant furin converts A110 to A60 and A30 without leaving a trace of the half-cleaved A70, even though the molecules where one β A subunit is more cleavable than the other were clearly present in that sample (Fig. 2A, compare lanes 1 and 2 in panel ii for wild-type β A, and also lane 2 in panel iii for the R310A mutant)
- ii. endogenous furin is also essential to further process A70 that is formed by endogenous Activin-A in human melanoma cell lines, ruling out an overexpression artifact.
- iii. A60 does not differ from A70 in its composition (as determined by LC-MS analysis and on reducing gels), but only in that the second β A subunit that is resistant to KLK8 underwent furin cleavage.
- iv. both A60 and A70 can specifically bind activin type II receptor, and only A70, but not uncleaved proActivin-A (A110) can signal in co-cultured HepG2 reporter cells, arguing against aberrant folding of either of these isoforms.

- v. a disulfide bond between the mature region and cysteine C35 or, if C35 is mutated, C38 in the prodomain persists even in A60 and is essential to stabilize it.
- vi. mutations that prevent disulfide bond formation between the mature region of Activin-A and its prodomain enable even uncleaved precursor to precociously bind cognate type II receptors independently of furin. As discussed in the co-submitted manuscript, this strongly suggests that the default connectivity of cysteines in proActivin-A prior to furin cleavage differs from the one inferred from crystal structures in which C35 and C38 were removed to obtain these crystals.

In short (and as discussed in depth in the co-submitted manuscript), cells expressing endogenous furin directly convert A70 to A60 by simply cleaving the remaining β A subunit that was resistant to KLK8, but then the prodomain of that subunit tends to remain covalently linked to the mature region via a disulfide bond of cysteine C35 with C322.

4. In Fig. 1E, the rationale for selecting specific siRNAs to validate the A70/A110 ratio is not clear.

We would like to thank the reviewer for pointing this out. We now clarified it in the text that our initial selection of 12 candidates was determined by choosing a cutoff of at least 5-fold reduction in Activin-A signaling activity as a threshold (by mistake the original text referred to a 10-fold reduction, but with that more stringent cut-off, only 5 candidates would have been retained, rather than the 12 (plus Klk6) described here. The revised text on p6 (line 4) now also clarifies that we further shortlisted five top candidates based on available literature about their substrate specificity, tissue distribution, or known links to diseases. Klk6 was included as it belongs to the same family as the top candidate (Klk8). However, siKlk6 did not decrease A70 formation, and treating proActivin-A with recombinant KLK6 did not generate hemicleaved Activin-A.

5. The YUMM3.3- β A group shows considerable variability within the dataset in Fig. 3F. It is recommended to investigate the cause of this variability and ensure experimental consistency.

Indeed, investigating possible causes of this variability proved to be informative: First, analysis of the RT-qPCR products by gel electrophoresis at the endpoint confirmed that *Klk8* band was amplified in the expected samples, but close to background in controls without retrotranscriptase (-RT) or absent with only water (see image below, first panel from left). Secondly, a comparison using three different primer pairs each confirmed a similar increase of *Klk8* expression in YUMM3.3- β A compared to Ctrl cells. Thirdly, examination of our records indicated that cell density may play a role since in the replicate with the highest *Klk8* mRNA levels, the cells reached nearly 100% confluence, compared to only around 75% in the other experiments. To directly test if *Klk8* expression levels are modulated by cell density, we seeded YUMM3.3- β A cells at increasing cell density, followed after 48 hrs by RT-qPCR analysis. The results from two independent experiments show that *Klk8* mRNA levels increased in a non-linear way with increasing cell density:

Prompted by these observations, we repeated the whole experiment in Fig. 3F two more times. As the results confirmed our initial conclusion, these new data were merged with the previous replicates in a revised Fig. 3F. In addition, the revised figure now also includes a picture of RT-PCR products and of the negative controls. By contrast, the previous replicate experiment from cells that were too dense was removed as an outlier. Finally, to further test the role of Activin-A signaling, the YUMM3.3- β A were treated with the ALK4 inhibitor SB 431542, both in three of our previous experiments and now also in the two new repeats. As shown in the revised figure 3F, administration of 10 μ M SB 431542 reduced *Klk8* expression in YUMM3.3- β A cells to baseline levels, further corroborating the specificity of the RT-qPCR analysis.

6. Attention should be paid to the labeling of protein bands for A110, A70 and A30, especially in figures such as Fig. S1B and S1F, where the position marked for A30 appears to be below 25kDa, which could confuse readers.

At the acrylamide concentration of 9%, A30 always migrated below the 25 kDa marker in every blot, not just in S1B and S1F. However, thanks to this sharp sighted observation of the reviewer, we noticed that the 25 kDa in source data file of Fig. S1D pointed by mistake at the 15 kDa marker at the front of the gel, instead of the actual 25 kDa marker. This error has been corrected in the revised version of this source data figure.

7. It is advisable to use asterisks to denote levels of statistical significance between groups to enhance clarity and readability.

To reduce the "cluttering", we now removed the numbers in cases where relevant differences reached statistical significance. In a few instances where numbers were retained, they serve to indicate how far those comparisons were from reaching significance.

Reviewer #2 (Remarks to the Author):

The manuscript by Buillard et al describes a novel mechanism for proteolytic activation of activin A (usually spelled without a hyphen) from its pro-forms by kallikrein-8. Activin A, like other TGF- β family proteins, is produced in precursor form which is processed at least by one proteolytic enzyme, typically furin or furin-like proprotein convertase, in a polybasic site separating the pro- and mature domains. Authors have previously shown that knocking out furin results in substantial suppression of proteolytic processing of pro-activin A. However, some processing does occur, to a partially (hemi) cleaved form, but the identity of the protease responsible for this was not known. This current manuscript seeks to address that.

The authors performed an siRNA screen of known protease genes which suggested that the likely candidate responsible for pro-activin A processing is kallikrein-8 and most of the manuscript deals with verification of this finding.

1. While identification of the alternative processing of pro-activin A is interesting, the manuscript fails fully to address the biological relevance of this processing, or to propose a mechanism to explain why kallikrein-8 would cleave just one of the two linkers between the pro- and mature domains in the dimeric growth factor.

The biological relevance was tested even by two independent approaches, first by demonstrating that *Klk8* RNAi *in vivo* specifically inhibited the growth advantage of β A but not of control tumors. Secondly, we show that a mutation that abolished cleavage by KLK8 *ex vivo* and in cell-free assays also abolished the activin-induced tumor growth, further corroborating that cleavage of other potential *Klk8* substrates was not relevant for the observed protective effect.

The mechanism how one β A subunit could become more susceptible to cleavage than the other is investigated and discussed in the co-submitted manuscript. The conclusion is that allosteric disulfides normally limit KLK8 access to the RRRRR motif and that binding of one β A subunit to an endogenous factor is rate-limiting for KLK8 cleavage of that subunit, or prohibitive on the other. Indeed, steric hindrance at a furin recognition site has been observed previously during Notch precursor processing, which resulted from the association of Notch with the regulatory protein Botch (Chi et al., 2012). The revised Discussion now cites this study (p13, 1st paragraph).

We are not sure how much of this finding is relevant in biology and not simply observed in experimental systems in which activin A is over-expressed.

The reviewers are correct that we here used mainly model cell lines that we engineered to ectopically express β A as a transgene. However, for the following reasons, we are confident that we are not studying an overexpression artifact:

- i. Most importantly, the hemicleaved form A70 and in some cases A60 can also be observed at endogenous *INHBA* expression levels in human cancer cell lines. An example was provided in Fig. 1C of the co-submitted manuscript, showing that endogenous A70 in human C8161 accumulates at the expense of A60 and A30 after stabilization by furin RNAi. However, in melanoma cells, these intermediates are harder to detect at endogenous expression levels since further processing by furin and subsequent turnover of the mature forms A30 and A60 is apparently faster than A70 production, a phenomenon that reminded us of our analysis of the effect of furin-mediated processing on Nodal turnover (Le Good et al., 2005; Blanchet et al., 2008). Since our co-submitted manuscript has now been reviewed by Communications Biology instead of Nat. Comm., we now revised figure 1 of the present manuscript, first by adding a Western blot from SCC-12 and SCC-13 cells where both A60 and A70 can be clearly seen in conditioned medium at endogenous Activin-A expression levels (new Fig. 1B).

In addition, we now show that treatment with the protease inhibitor dec-RVKR-cmk (CMK) blocks A70 and A60 formation both in the SCC cell lines and in human melanoma cell lines, similar to what is observed with overexpressed Activin-A (new Fig. 1C). The effect of CMK treatment further confirms that these bands are specific Activin-A cleavage products. Note that these cell lines also express furin (and possibly related PCs), and the CMK concentration required to block A70 formation is therefore higher than in the furin-deficient FurKO- β A cells, as expected.

- ii. We previously validated that the tumor-promoting function of immunosuppressive paracrine Activin-A signaling that was initially discovered in β A overexpressing tumors recapitulates a function of endogenous Activin-A signaling in syngeneic grafts of iBIP2 mouse melanoma cells (Donovan et al., 2017; Pinjusic et al., 2022). Although not shown in the present manuscript, iBIP2 cells also secrete A70, and we applied for an animal license to use that model in future studies to further investigate the function of additional PCLP candidates.
- iii. Here and in our previous study, we measured Activin-A levels in the circulation of tumor-bearing mice to be in the range of 35 ng/ml (this study) to 40 ng/ml (Pinjusic et al., 2022). These concentrations are still below those released into the plasma by tumors in *Inha*^{-/-} mice (Matzuk et al., 1994). We now added this information in our Results section on p11 (1st paragraph). In human, the levels of circulating of Activin-A are often below 10 ng/mL, but in colorectal cancer patients it reached 17.6 ng/mL (Loumaye et al., 2017). Even though it is unknown whether the accumulation in plasma directly correlates with expression levels in a given tumor or whether it is regulated

differently in mice compared to humans, we think these numbers show that the Activin-A levels analyzed here are not in a wildly unphysiological range.

- iv. As shown in figure S1C of the co-submitted manuscript, our engineered cell lines were carefully selected to express comparable levels of βA mRNA. Side-by-side comparisons indicate that the steady state concentration of their secreted Activin-A protein is not more than 10-fold above the concentration in human melanoma cell lines (see image below).

To study the regulation of precursor processing, it was both reasonable and essential to use a system where the detection of short-lived processing intermediates is sufficiently robust to obtain high quality data. Another consideration is that we cannot use cell lines that may express additional inhibin genes that could confound any interpretation by forming heterodimers.

We did not try to test the physiological role of KLK8 in regulating Activin-A signaling in human tumors, also because *in vivo* analysis of the effect on Activin-A function is only possible in syngeneic grafts of

mouse cells in immunocompetent hosts. Nevertheless, after confirming that the seed

sequences of our siKlk8 sequences are quite similar to human KLK8 (in the image to the right, differences are highlighted by red crosses), we now tried to use them to deplete KLK8 alone or together with furin in A375 human melanoma cells *ex vivo*. KLK8 mRNA was depleted by about 40%, which is slightly less efficient than in murine cell lines, where the depletion reached 50-60% (Fig. S11). KLK8 depletion alone led to a significant 2-fold increase in A70 by endogenous Activin-A, consistent with our model that KLK8 contributes to the cleavage of only one βA subunit of Activin-A.

However, the combination of siFURIN/Klk8 failed to completely inhibit A110 processing, suggesting that furin likely acts redundantly with one or several other PCs in this cell line. Therefore, to obtain a more definitive answer, one would have to completely delete furin by CRISPR editing alone and together with any other co-expressed PC. However, this would be hard to justify, given that we have gone already through the whole nine yards to do this in the mouse model where the impact on immune evasion by Activin-A in melanoma was testable (and confirmed) *in vivo*.

2. Some of our scepticism arises from the authors' own interpretation of their data. For example, in the siRNA experiments even control cells with non-targeting siRNA (siNT) fail to process activin A correctly and this treatment reduced activin activity by 50%. If the latter was really due to "edge effects" as authors suggest, did they run these samples in the middle of the plate to confirm it? Indeed, if there was a significant edge effect (which could be mitigated), we hope the whole experiment was repeated with samples in different places.

First, we would like to point out that the conclusion of our screen in no way depends on siNT or on the interpretation of its effect: Since siNT (a randomly chosen, commercial control siRNA that is marketed *by the vendor* as "non-targeting") evidently had some effect (be it a border effect or whatever), we decided to not rely on this or any other random siRNA to determine a baseline. Instead, the baseline was determined using the average of all siRNAs of non-candidates, the vast majority of which had no effect on CAGA-Luc activity.

We also did not claim that the edge effect was the sole reason. We instead cautiously stated that the effect *correlated* with the position of siNT at the edge. For the siRNA screen, all cells transfected with siNT were indeed located at the border in the first two columns on the left. Since this screen contained no less than fourteen 96-well plates (each siRNA plate in duplicate), we only repeated plate 1 and found that the siNT effect was reproducible, but at that stage we did not consider if it was due to the position and therefore did not change the position. In this repeat experiment, siNT diminished activin signaling by 20% (see graph to the right). Repeating the whole screen would have been very difficult to justify the effort and cost, given that our screen did not depend on this or any other NT siRNA.

Moreover, during the validation of hits, siCyto and siNT were each time included again as controls. These experiments (in 24-well plates) showed that siNT consistently diminished the amount of total Activin-A. This can be clearly seen in all of our Western blots (see for example Fig. 1E and 1F). However, rather than invalidating any of our conclusions, this observation actually supports a role of KLK8, because it confirmed that a mere decrease of total Activin-A secretion was not sufficient to also bring down the A70/A110 ratio below our threshold of 5-fold reduction.

Little information was provided about cell viability as well in these early experiments, which was an issue later. What was the effect of KLK8 siRNA treatment on cell survival. Perhaps authors can provide data for that.

The effect of the *Klk8* siRNA pool on cell viability in 96-well plates was actually already quantified (Fig. S1G). In our view, it was important to analyze and show this since the effect was clearly not negligible. Indeed, already during the RNAi screen, each well was observed in the microscope, and wells containing fewer cells or debris from dead cells were marked accordingly before adding the reporter cells. Around 30% of the wells had less cells. As shown in the picture below, aside from such debris (arrowheads), the remaining cells in the siKlk8 wells looked normal and indistinguishable from cells transfected (for example) with siFurin:

Furthermore, to account for differences in cell numbers, the CAGA-Luc signals observed in HepG2.α1PDX reporter cells after RNAi in all wells were normalized to the total Activin-A secreted by viable cells, including uncleaved precursor. As described in the Methods section, total Activin-A was measured using control reporter cells that did not express the furin inhibitor α1-PDX. To ensure that these measurements were within the dynamic range of the assay and did not saturate CAGA-Luc, only 20 μL (from a total volume of 200 μL) of the conditioned medium was added to HepG2 CAGA-Luc cells. As shown in the figure to the right, these measurements confirmed that total Activin-A in wells treated with siKlk8 was not significantly depleted until the endpoint, confirming that enough cells remained alive, in contrast to siCyto-treated where the signaling by residual Activin-A was drastically diminished after knockdown of *Klk8* or of any other candidates. These data confirm that our strategy allowed to distinguish genuine candidates from false positive toxic siRNA pools.

3. KLK8 was also not the only gene (or pool for that matter) that had significant effect on proactivin A processing. What happened to Adamts10? Its inhibition had a similar effect to KLK8, but it seems to have been dropped off as a potential candidate without validation. Will there be a separate publication on that?

Although *Klk8* was the top candidate of the screen, it was indeed not our only candidate of interest. Similar to the experiment performed to confirm the effect of *Klk8* depletion (Fig. 1F), the five final selected candidates including *Adamts10* were tested by transfection of the four individual siRNAs present in the original pool. However, in contrast to *Klk8* siRNAs, only two of the four *Adamts10* siRNAs impaired A70 formation, and less completely than *Klk8* RNAi if the signals for residual A70 protein and of its signaling activity were normalized to total Activin-A. Furthermore, we now state on p12 (2nd paragraph) of the revised Discussion that *Adamts10* is not commercially available as a recombinant protein. Its functional analysis is indeed still ongoing. Here, we instead prioritized KLK8.

4. The authors also conducted a validation experiment in which cell culture supernatants containing activin A were treated with recombinant kallikrein-8. Processing by kallikrein-8 required acidification of the sample to a pH of at least 6.5, but data were shown for pH 2. At this low pH, we struggle to see how this processing could be physiologically relevant or even that it is not an artefact resulting from pH-driven changes in protein structure. Acidification is routinely used to release TGF-β from its latent form, but reduction in processing by furin could be simply an indication that treatment is causing activin A to behave non-natively and not a demonstration of any specific mechanism.

Trying to at least transiently "induce a pH-driven change in protein structure" (or in protein complexes) was indeed the stated purpose of this experiment: As described in the Methods section, recombinant KLK8 was only added after setting the pH back to neutral, and rKLK8 is already active since it was preincubated for 30 min with lysyl-endopeptidase. A control of acidified SN without the addition of rKLK8 was always performed in each experiment. This control shows that acidification alone was not enough to induce proActivin-A cleavage, ruling out the possibility of being an artifact simply resulting from changes in protein structure. In addition, as pointed out by the reviewers, to assess if this effect of acidification of proActivin-A could be physiologically relevant, we confirmed that even a mildly acidic pH is already sufficient. However, it is correct that our experiments initially used the non-physiological pH 2 since such transient acidification can activate latent TGF-β, and because the differential susceptibility of the two βA subunits to undergo hemicleavage implied that one subunit

adopts a different conformation than the other, most likely because of an associated interacting factor. As discussed in the co-submitted manuscript, possible candidates include an LTBP-like factor, since it is known that also the conformation of latent TGF- β in a complex with LTBP (or with the unrelated protein GARP) must be asymmetric due to the known stoichiometry of only one LTBP molecule per TGF- β dimer. Exactly for the reasons mentioned by the reviewer, we asked if the cleavage observed at pH 2 also occurs at a more physiological pH. The pH titration experiment shown in **Fig. S2B** showed that even a mild transient acidification at pH 6.5 was sufficient to induce cleavage by rKLK8.

As discussed in the co-submitted paper, our analysis of cysteine mutant forms of Activin-A indicates that the function of an LTBP-like transporter protein is likely responsible for modulating the folding of allosteric disulfide bonds in the associated β A subunit of proActivin-A. By default, we find that C35 in the prodomain is linked to C314 in the mature region to prevent precocious ActR-II binding during secretion, but that in a fraction of proActivin-A, a rate-limiting interacting factor redirects the C35 of one β A subunit to instead pair with C322. We propose that while this alternative conformation favors KLK8 access, cleavage of the first subunit in turn facilitates the release from that associated factor. While this is not the only possible model, it is the simplest one and therefore the one that we propose in our co-submitted manuscript that is now under revision at Communications Biology where it received favorable reviews. We also presented this model at the 2024 FASEB conference on the TGF- β superfamily, where it raised considerable interest both among the structure biologists in the field, and among researchers who observed analogous processing intermediates of GDF8/myostatin that contain the same cysteines at the N-termini of the prodomain and mature region as activins.

5. One of the key experiments underlying the story is an siRNA screen against 542 proteases. This was performed in a cell line overexpressing activin A, using lentiviral transgene. It is not clear what level the expression of activin A is in these cells, how it relates to physiological levels and how overexpression may affect activin A processing. Some of the interpretations are also partly dependent on a parallel manuscript by the same authors on alternative forms of activin A. The authors discuss the non-canonical 58 kDa (A60, see later comment on nomenclature) cleavage product, but the relevance of this is not clear.

Regarding overexpression, please see our answer to questions 1 and 6. In A60, the prodomain of one subunit remained disulfide-linked to the mature region even after furin cleavage. We think it is only "non-canonical" in the sense that the mechanism underlying this phenomenon still remained to be discovered. The characterization of A60 and its potential relevance are further discussed in the co-submitted manuscript. Since activin-induced tumor growth does not correlate with steady state levels of A30, one intriguing (and in our view the most likely) scenario is that the paracrine signaling mediating tumor immune evasion involves A60. Indeed, owing to covalent linkage of at least one of its cleaved prodomain subunits, A60 can be expected to differ in terms of its bioavailability in different cell types of the tumor microenvironment.

However, please note that the present manuscript is not about A60 or its function, but about identifying a mechanism mediating furin-independent hemicleavage of proActivin-A using B16-F1 cells as a previously validated model system, and about further testing a role for KLK8 in this process *in vitro* and *in vivo*, together with an analysis of *KLK8* expression in public cancer databases.

6. It is well known that overexpression of these proteins in mammalian cells can result in high molecular weight aggregates and secretion of partially processed forms. The latter can be corrected by furin overexpression, suggesting that the intrinsic capacity of cellular machinery is limited for proprotein processing.

As mentioned in our answer to point 1, Activin-A also forms A70 and A60 at endogenous expression levels in human skin cancer cell lines (new Fig. 1B and 1C), ruling out that such alternative forms are an overexpression artifact. Furthermore, contrary to the hypothesis of the reviewer that the endogenous processing machinery might have been overwhelmed, the partially processed A70 generated by our 10-fold overexpressed Activin-A precursor was as efficiently and completely converted to mature forms by *endogenous* furin as the endogenous A70 in human skin cancer cell lines. Neither this study nor the accompanying manuscript make any use of furin overexpression.

"...of these proteins": Which ones did the reviewers have in mind? For the distantly related activin receptor ligand Nodal, our lab previously reported that it forms orderly high molecular weight multimers of dimers (as determined by gel filtration). Although these are too big to even enter non-reducing SDS PAGE gels, they appear to be correctly folded, as evidenced by the fact that they do not hinder at all either precursor processing or signaling (Fuerer et al., 2014). We are not aware of any reports that activins or other TGF- β family members form similar multimers, or that they are generally prone to aggregate either on non-reducing gels or on gel filtration columns, let alone that such hypothetical "aggregation" would alter Activin-A precursor processing. As discussed in the co-submitted manuscript, a disulfide linkage of the prodomain to its mature region has once been interpreted as being prohibitive for furin cleavage of TGF- β many years ago (Gentry et al., 1988). However, that study cannot rule out that the S-S bond instead stabilized a latent complex, a scenario which in our view is more likely. In any case, our finding that endogenous furin in *Furin* WT B16F1- β A cells completely processed even our overexpressed Activin-A shows beyond any doubt that an analogous S-S linkage definitely does *not* block furin cleavage of Activin-A.

Of note in this regard are the higher molecular weight bands which are marked as "ns" (non-specific) in some of the western blots. It would be appropriate to confirm that they are truly result of non-specific antibody reactivity and NOT activins in aggregated state.

The three only instances where a HMW band is marked "non-specific" (ns) all refer to blots of plasma samples (Figs. 5C, 5F, and 6F). This band is labelled "non-specific" because it is unrelated in size to Activin-A and was equally present in plasma of mice that bear only Activin-A negative Ctrl tumors (Fig. 6F, lanes 1 and 2). For further comments, please see also our answer to the similar question of reviewer #2 in point 8.

7. The authors then use mutagenesis of the polybasic furin site (overlapping with putative kallikrein recognition/cleavage site) to interrogate the key residues required for cleavage by kallikrein-8. We are not convinced that the mutagenesis experiments conducted provide conclusive evidence that the R310 specifically favours cleavage by kallikrein-8 as described.

The authors first show that mutation of R310 to alanine abolishes kallikrein-8 processing of pro-activin A. They then speculate this is possibly due to changed sequence context of the processing site and examine this hypothesis by moving the rest of the sequence up by one, resulting in sequence RRRRALE (R310A mutation, elimination of "native" Gly from

RRRRRGLE). Why didn't the authors just remove the 5th arginine, rather than retain the mutated alanine and remove the natural glycine? This change is not very large, agreed, but at the same time it does not fully replicate the sequence context of the native cleavage site.

The effect of "eliminating" the 5th arginine ($\Delta R310$) that the reviewer asks for was in fact analyzed in figure 2G, side-by-side with the R310A and the RALE mutant. Of note, this simple deletion of the 5th arginine similarly restored Activin-A cleavage by KLK8, similar to what we also observed with the RALE mutant. This finding and its interpretation were also mentioned (again) in the Discussion where we stated: *"Importantly, however, both the formation of hemicleaved A70 by KLK8, as well as its further maturation by furin, were fully rescued if R310 was simply deleted, or if the RA₃₁₀GLE sequence in R310A mutant proActivin-A was mutated to RA₃₁₀LE. Taken together, these findings strongly support our interpretation that conformational constraints regulate the positioning of scissile bonds in the penta-arginine motif of full-length proActivin-A, but not in the identical sequence of a flexible peptide."* (p13, last lines)

The resulting discussions on structural context of the site are simply nonsensical – with the exception of MSTN (where the entire furin site is seen in the crystal structure), the furin site is poorly structured (flexible) and not resolved in other TGF- β structures.

The available structures come from proteins that were either folded *in vitro* or purified from cell conditioned media, and never from cell lysates. After explaining which data in the present study question the notion of a freely flexible furin site, our Discussion concluded on p14 (1st paragraph): *"...Taken together, these findings strongly support our interpretation that conformational constraints regulate the positioning of scissile bonds in the penta-arginine motif of full-length proActivin-A, but not in the identical sequence of a flexible peptide."* As discussed there, the rescue of cleavage by the $\Delta R310$ mutation that reviewers 2/3 apparently overlooked was indeed one of our key results that influenced our reasoning. While studying the role of furin and related proteases in Nodal, BMP, and then Activin-A processing, we too initially assumed for >20 years that their cleavage sites should all be flexible. However, this model could not explain the observations about Activin-A processing. Our new model of precursor processing predicts that the configuration of disulfide bonds of cysteines C314 and C322 adjacent to the furin motif differs from those in the proActivin-A crystals. For as discussed in the co-submitted paper, to obtain those crystals, the authors had to remove the cysteines C35 and C38, and they rightly pointed out that the resulting structure failed to explain why ActR-II receptors cannot extract mature Activin-A from its complex with prodomain independently of furin cleavage. Our co-submitted manuscript provides an explanation for this conundrum by demonstrating that precocious access of ActR-II to uncleaved proActivin-A is prohibited by cysteines C314 and C322 which we found to be responsible for tethering uncleaved mature region to the prodomain. As mentioned above in our answer to point 1 of reviewer #2, we concluded that such allosteric disulfides normally limit the flexibility of the nearby RRRRR motif. A role for other additional factors also cannot be excluded.

How to best cite our complementary study will depend on how its publication will be coordinated by Nature press: It received favorable reviews, and we are about to resubmit it in parallel to this one. However, to further address the point raised by reviewers 2/3 that the cleavage site is generally thought to be flexible, we added to the Discussion the following sentence (at the top of p13): "Why KLK8 access is limited to one β A subunit will be discussed in a separate study where we analyzed the potential influence of adjacent disulfide bonds on the penta-arginine motif."

8. When it comes to biological validation of the possible kallikrein-8 processing of pro-activin A, the work relies again on overexpression of activin A to see the effect – cells without

overexpression do not show difference in tumour growth. Is this physiologically significant?

As mentioned in our answer to point 5, the role for endogenous Activin-A in promoting immune evasion and immunotherapy resistance in melanoma has been previously validated in syngeneic iBIP2 grafts. That same work showed that *INHBA* is also among the top upregulated genes in immune checkpoint therapy-resistant melanoma patients. Other authors in the meantime have independently confirmed the finding that Activin-A in human melanoma and associated immune suppressive phenotype of macrophages correlate with poor prognosis (Gutiérrez-Seijo et al., 2022; Pich-Bavastro et al., 2023). Therefore, the clinical relevance of Activin-A is not the theme of the present study. Instead, we here focused on evaluating a role for *KLK8*.

Given the frequent expression of *INHBA* in human melanoma, targeting *KLK8* may benefit many patients. We expect a similar role in other cancer types where *KLK8* is expressed. In addition, our analysis of human databases shows that *KLK8* expression in human melanoma correlates with the disease stage at the transition to metastasis, which suggests that targeting *KLK8* warrants consideration as a therapeutic approach in this and possibly other cancers that coexpress *INHBA* and *KLK8*. In addition, high *KLK8* expression correlates with reduced overall survival in many cancer types, suggesting that targeting *KLK8* may also be beneficial outside of an Activin-A context. In our B16-F1 model it is not the case, but these cells only express *KLK8* at relatively low levels (**Fig. S1G**).

In these experiments (Fig 5 and 6) the western blots in particular could do with better labelling as sample numbers are not identical in all cases for tumour and serum samples.

We are grateful to the reviewers for pointing this out. We now added the number of each mouse on top of each lane of the Western blots to enhance the clarity and to more easily recognize which tumor sample corresponds to which plasma.

Also, why is the western blot in Figure 6 suddenly showing mostly proteins at much higher molecular weight, well beyond 110 kDa species of activin A, compared to other experiments?

As mentioned above (point 6), this blot in figure 6 (panel F) shows plasma samples. As stated on p11 of the Results section (end of 1st paragraph), the conclusion of Fig. 6F is that even if we block furin cleavage (mS1 mutant) or the hemicleavage by *KLK8* (R310A mutant), no unprocessed or hemicleaved Activin-A increased in the circulation (consistent with the notion in the literature that the prodomain can "anchor" these species at the source by binding e.g. to specific ECM proteins). However, since it was not known *a priori* whether these or even larger forms of Activin-A nonetheless accumulate also in the circulation, we deliberately didn't crop the high molecular weight region. Fig. 6F shows that the same non-specific HMW band was present and equally abundant in Ctrl tumors that are *not* expressing Activin-A, which is why it is safe to conclude that it is indeed non-specific.

The two other blots of plasma samples (Figs. 5C and 5F) also include this region above 110 kDa. Although this was not a place to comment about it, the result (and corresponding blots of plasma from Ctrl tumor-bearing mice that were not included in Fig. 5 to avoid cluttering) confirm that this non-specific HMW band was consistently observed across multiple experiments.

9. Finally, we also wanted to address authors' use of abbreviations, especially for the different forms of activin A. They observe uncleaved 110 kDa form, hemi-cleaved 66 kDa form, alternative hemi-cleaved form at 58 kDa and the mature growth factor of 26 kDa (which by the way runs below 25 kDa on all of the gels!) in non-reduced SDS-PAGE/western blot analyses. Why are the names for the different forms different from the observed molecular weights? Why is 66 kDa form A70, 58 kDa A60 and mature domain as A30 (why not just call

mature activin A as that, like is customary). This adds unnecessary level of convolution and certainly follows no expected standard. It simply makes no sense to round up these numbers.

Under the subheading Western blot analysis in the Materials & Methods section, we now added a sentence explaining the logic of the A30/A70 nomenclature by stating: "In accordance with Ginefra et al. ¹⁵, mature Activin-A was labelled A30 by rounding up (as in the case of A70) the predicted actual size (2x13 + 2 kDa) of the myc-tagged dimer, rather than according to an apparent molecular weight that varies depending on the acrylamide concentration." Indeed, to credit those who initially described hemicleaved Activin-A as a 70 kDa protein (Huylebroeck et al., 1990; Mason et al., 1996), we decided already in Ginefra et al., 2018 to simply call it "A70". Compared to the apparent molecular weight of 66 kDa that we observed for this glycosylated protein under the specific conditions examined, this name involved a rounding up, which is why we then chose to similarly "round" the names of canonical mature form with its myc tag, and of A60 (migrating at 58 kDa) – a species that we initially expected to perhaps arise by further cleavage of A70, until further analysis argued against this hypothesis: In our view, for the sake of de-"convolution" and readability, this simplification was the best compromise that we could find.

Indeed, until we arrived at this solution, choosing names for these isoforms gave us considerable headache since the activin field has not come up with a unified simple nomenclature to cope with the complexity of cleaved and uncleaved forms of homo and heterodimers. The initial names proposed for mature Activin-A and its precursor, respectively, were $(\beta A)_2$, $(PRO\beta A)_2$, and $PRO\beta A-\beta A$ for hemicleaved form (Huylebroeck et al., 1990). By contrast, Mason et al. (1996) subsequently used $pro\beta A$ to distinguish the prodomain from cleaved mature region (βA), and they called hemicleaved $pro\beta A-\beta A/\beta A$. Speaking of "convolution", we think that neither suggestions of these "fathers" of Activin-A hemicleavage are sufficiently readable. They also require too much space in multi-panel figures, and they were not amenable to an intuitive and legible way to then readily distinguish between a hemicleaved form (e.g. $PRO\beta A-\beta A$) and its fully cleaved derivative with a disulfide-linked prodomain still attached (here called A60) (**Fig. 1A**). Furthermore, βA is a much needed and in our view also more logical abbreviation for the $INH\beta A$ transgene and the corresponding full-length polypeptide. Apparently many other authors also find the canonical name "activin A" impractical. This is true especially for *our* line of work with its focus on factors that "activate activin A by removing the activin A prodomain to induce binding of mature activin A to activin type IIA receptor": We find that the combination of activin (lower case) with the capitalized (lonely) A's confuses non-specialist readers, not to mention the increased word count. Consistent with our previous publications, we therefore call it "Activin-A". For the sake of clarity, the revised manuscript now introduces the original name "activin A" in the Introduction (consistent with how we did it also in the co-submitted manuscript).

Given our finding that A60 is not less cleaved than A30, the term "mature Activin-A" (which by definition refers to cleavage, not to a size) is clearly inadequate to distinguish these two mature forms from each other. Therefore, we decided that in analogy to the existing A70/A30 nomenclature, A60 would be the best possible short shrift for this fully cleaved A70 derivative, at least until those of us who have led the field of TGF- β /BMP processing can agree on a unified nomenclature, for example in a future review.

The manuscript is also otherwise a riot of unique abbreviations: SN for supernatant, DKO for furin/PC7 double knock-out, FurKO for furin knock-out, ...

Supernatant (SN) is synonymous to conditioned medium (CM) and very common in a large body of literature. In our view, SN is preferable to CM because it is less easily confused with the co-occurring (and equally standard) abbreviation "CL" for cell lysates. DKO for double

knockout is very common as well: It was thus introduced already in Ginefra et al. specifically to distinguish CRISPR-engineered *Furin*; *Pcsk7* double knockout cells from single mutant *Furin* KO (FurKO) cell lines. Moreover, since the original acronym of the gene encoding furin was FUR for "FES upstream region", FurKO was and still is the most appropriate name for the first cell line reported in the literature where furin has been knocked out.

PCLP for PC-like protease (google search comes up with "podocalyxin-like protein")... There is a time and place for abbreviations and saving space, but it cannot be at the expense of readability and accuracy.

While we fully agree with the reviewers that abbreviations should be used with care, any new short abbreviation will be found by a Google search in some obscure places. In the case of PCLP, we think it is our best choice, not least because it is clearly distinct from the gene acronym of the said gene, which is PODXL, not PCLP. PubMed contains only 2 papers in poorly known journals where PODXL was called PCLP (as compared to >200 papers that properly refer to PODXL). Furthermore, since we use "PCLP" to refer to an activity rather than to any specific *PCLP* gene (in italics), a possible confusion is improbable.

There is also some inconsistency in using gene and protein names in the text.

We apologize that our zeal for clarity in this regard apparently still left some ambiguity: After additional rounds of checking, the two or three instances where we omitted italics have been corrected.

Overall, we are on the fence about this manuscript. There is a significant amount of work in this manuscript which is mostly of high quality and kallikrein-8 cleavage has been shown convincingly in this experimental setup. However, we are not convinced about the significance of these findings either in normal biological context or in disease

We would like to sincerely thank reviewers 2/3 for taking the time to read and kindly review our manuscript, and for their various constructive criticisms: Since we believe that we were able to address all comments, we hope that they will now find our revised manuscript to be sufficiently convincing.

Reviewer #3 (Remarks to the Author):

Reviewer #4 (Remarks to the Author):

I find this a comprehensive study showing that KLK8 participates in the stepwise cleavage of Activin-A precursor protein which seems to be involved in the progression of melanoma tumours.

I have two comments that I think may improve the quality of the manuscript:

1. Figure 4C: The authors claim they show representative figures of crystal violet staining of cells at the experimental endpoint. As I understand the staple diagrams, there are stronger signals in the lower panel. When I look at the photographs included, I can see more spots indicating stained cells in the upper panel, and no spots at all in the two pictures to the right in the lower panel.

We are most grateful to the reviewer for noticing this inconsistency! The crystal violet areas in the main figure were accidentally not yet updated to match the same regions as the ones highlighted for a final version in the source data which better represent the overall quantification.

2. Discussion section “KLK8 can convert proActivin-A to hemicleaved form”: I agree with the authors concerning the overall conclusions drawn in the discussion, but I have some issues concerning how some references are used in the context as exemplified below:

- Line 411 ref (34) is misleading in this content. In this paper, the inhibition of KLK5, KLK7 and KLK14 by LEKTI fragments are studied, and it is shown that this inhibition is pH-dependent which is referred to in the text. But KLK8 is not included in that study. In Eissa et al (ref 32) it is mentioned that KLK8 is not inhibited by any LEKTI fragments.

Although KLK8 was not mentioned in our sentence, we agree that this could be misleading and therefore rephrased our sentence to specifically mention *which kallikreins have been shown to be affected by acidification* (p13, 1st paragraph): *"Acidification in vitro has been shown to overcome binding of several kallikreins to inhibitory proteins, including KLK5, KLK7, and KLK14, raising the possibility of a similar effect on KLK8³⁸."*

- Line 417-418 it is stated “...a pH-regulated proteolytic cascade initiated by KLK5 is important for the zymogenic maturation of KLK8” and the ref for this is Eissa et al (32). I agree with the authors that this may be the case, but in the paper of Eissa et al they show that KLK8 is active also during acidic conditions, but all activation experiments were performed at basic conditions. The idea of a pH regulated proteolytic cascade initiated by KLK5 was initially published in Brattsand et al, *J Invest Dermatol* 124: 198-2003, 2005, where it was shown that the activation rate of proKLK7 by KLK5 peaked at pH 5,5 although the pH optimum for the KLK5 activity towards chromogenic substrates were slightly basic as for KLK8. Therefore, it could be hypothesized that the activation of proKLK8 by KLK5 could be affected by pH although it has not yet been published to my knowledge.

Indeed, this hypothesis is highly plausible, and it is one possible scenario of how an acidic pH might stimulate KLK8 activation *in vivo*. Therefore, Brattsand et al. (37) are now cited in the Discussion of our revised manuscript: *"For example, even though enzymatic KLK8 activity is optimal between pH 7 to 9³⁶, a low pH in vivo may facilitate its activation by KLK5, as shown previously for the zymogenic maturation of KLK7³⁷."* (p13, 1st paragraph). However, in our cell-free assay, the acidification step is transient and happens before the addition of rKLK8, so it would mainly affect the conformation of proActivin-A or of an interacting factor.

Bibliography

- Blanchet, M.H., J.A. Le Good, D. Mesnard, V. Oorschot, S. Baflast, G. Minchiotti, J. Klumperman, and D.B. Constam. 2008. Cripto recruits Furin and PACE4 and controls Nodal trafficking during proteolytic maturation. *EMBO J.* 27:2580–2591. doi:10.1038/emboj.2008.174.
- Cangkrama, M., M. Wietecha, and S. Werner. 2020. Wound Repair, Scar Formation, and Cancer: Converging on Activin. *Trends Mol. Med.* 26:1107–1117. doi:10.1016/j.molmed.2020.07.009.
- Chi, Z., J. Zhang, A. Tokunaga, M.M. Harraz, S.T. Byrne, A. Dolinko, J. Xu, S. Blackshaw, N. Gaiano, T.M. Dawson, and V.L. Dawson. 2012. Botch Promotes Neurogenesis by Antagonizing Notch. *Dev. Cell.* 22:707–720. doi:10.1016/j.devcel.2012.02.011.

- Donovan, P., O.A. Dubey, S. Kallioinen, K.W. Rogers, K. Muehlethaler, P. Müller, D. Rimoldi, and D.B. Constam. 2017. Paracrine Activin-A signaling promotes melanoma growth and metastasis through immune evasion. *J. Invest. Dermatol.* 137:2578–2587. doi:10.1016/j.jid.2017.07.845.
- Fuerer, C., M.C. Nostro, and D.B. Constam. 2014. Nodal-Gdf1 Heterodimers with Bound Prodomains Enable Serum-independent Nodal Signaling and Endoderm Differentiation. *J. Biol. Chem.* 289:17854–17871. doi:10.1074/jbc.M114.550301.
- Gentry, L.E., M.N. Lioubin, A.F. Purchio, and H. Marquardt. 1988. Molecular events in the processing of recombinant type 1 pre-pro-transforming growth factor beta to the mature polypeptide. *Mol. Cell. Biol.* 8:4162–4168. doi:10.1128/MCB.8.10.4162-4168.1988.
- Ginefra, P., B.G.H. Filippi, P. Donovan, S. Bessonard, and D.B. Constam. 2018. Compartment-specific biosensors reveal a complementary subcellular distribution of bioactive furin and PC7. *Cell Rep.* 22:2176–2189. doi:10.1016/j.celrep.2018.02.005.
- Gutiérrez-Seijo, A., E. García-Martínez, C. Barrio-Alonso, V. Parra-Blanco, J.A. Avilés-Izquierdo, P. Sánchez-Mateos, and R. Samaniego. 2022. Activin A Sustains the Metastatic Phenotype of Tumor-Associated Macrophages and Is a Prognostic Marker in Human Cutaneous Melanoma. *J. Invest. Dermatol.* 142:653-661.e2. doi:10.1016/j.jid.2021.07.179.
- Huylebroeck, D., K. Vannimmen, A. Waheed, K. Vonfigura, A. Marmenout, L. Fransen, P. Dewaele, J.M. Jaspar, P. Franchimont, H. Stunneberg, and H. Vanheeuverswijn. 1990. Expression and processing of the Activin-A erythroid differentiation factor precursor-a member of the transforming growth factor-beta superfamily. *Mol Endocrinol.* 4:1153–1165.
- Le Good, J.A., K. Joubin, A.J. Giraldez, N. Ben-Haim, S. Beck, Y. Chen, A.F. Schier, and D.B. Constam. 2005. Nodal stability determines signaling range. *Curr. Biol.* 15:31–36. doi:10.1016/j.cub.2004.12.062.
- Loumaye, A., M. de Barys, M. Nachit, P. Lause, A. van Maanen, P. Trefois, D. Gruson, and J.-P.P. Thissen. 2017. Circulating Activin A predicts survival in cancer patients. *J. Cachexia Sarcopenia Muscle.* 8:768–777. doi:10.1002/jcsm.12209.
- Matzuk, M.M., M.J. Finegold, J.P. Mather, L. Krummen, H. Lu, and A. Bradley. 1994. Development of cancer cachexia-like syndrome and adrenal tumors in inhibin-deficient mice. *Proc. Natl. Acad. Sci. U. S. A.* 91:8817–21. doi:10.1073/pnas.91.19.8817.
- Pich-Bavastro, C., L. Yerly, J. Di Domizio, S. Tissot-Renaud, M. Gilliet, and F. Kuonen. 2023. Activin A-mediated polarization of cancer-associated fibroblasts and macrophages confers resistance to checkpoint immunotherapy in skin cancer. *Clin. Cancer Res.* CCR-23-0219. doi:10.1158/1078-0432.CCR-23-0219.
- Pinjusic, K., O.A. Dubey, O. Egorova, S. Nassiri, E. Meylan, J. Faget, and D.B. Constam. 2022. Activin-A impairs CD8 T cell-mediated immunity and immune checkpoint therapy response in melanoma. *J. Immunother. Cancer.* 10:e004533. doi:10.1136/JITC-2022-004533.
- Yu, Q., and I. Stamenkovic. 2000. Cell surface-localized matrix metalloproteinase-9 proteolytically activates TGF- β and promotes tumor invasion and angiogenesis. *Genes Dev.* 14:163–176. doi:10.1101/GAD.14.2.163.

Point-by-point answers to reviewers

We would like to thank again all four reviewers for their time and constructive comments to help improve this manuscript, and for their unanimous recommendation to accept the revised manuscript for publication! For the remaining comments of reviewer #2, please find our point-by-point answers below.

Reviewer #1:

Authors have fully addressed my concerns. I suggest it for publication.

Reviewer #2-3 (reviewer #3 co-reviewed this manuscript with reviewer #2 who provided the listed report):

We would firstly like to give credit to authors for comprehensive replies to our (and other referees') comments. We also applaud them for re-running experiments to ensure consistency of data. We are still in two minds with the manuscript. There is no arguing with the technical quality and amount of work in this manuscript. The authors have left hardly any stone unturned to confirm their findings.

We would like to thank reviewers #2-3 for this kind summary statement, and for appreciating the effort already invested in this manuscript and its revision!

However, most of the queries which relate to the biological/medical significance (e.g. relation to human cancers) of this specific work on KLK8's role in activin signalling are still speculative to us.

Which answer(s) to any query remained "speculative" to them is difficult to see: Since reviewers #2-3 used no numbering, our answers grouped their specific questions into nine main points that were then further broken down individually. None of them even mentioned "cancers" or "human", or how the *role of KLK8* would have to be further validated. Instead, a doubt about "significance of our findings ...in disease" was only mentioned in one *general* final sentence. All *specific* queries related to this subject only questioned how we can be sure that neither the *effect of Activin-A on tumor growth* (point 8) nor the *existence* of the observed processing intermediates are merely overexpression artifacts (points 1, 4, 5, 6). Therefore, this is what our answers and new data in figure 1B-C addressed, including all other points mentioned below. Even now, reviewers #2-3 identified no omission *how else* we should have further tested "biological significance". To quote the reviewer's own comment about this revision: "They have left hardly any stone unturned to confirm their findings". We agree, because we believe that we were able to address not only most but *all* of the *actual* concrete queries, including those of reviewers #2-3.

One of the reasons for this relates to our original comment on the results of the protease siRNA screen where number of other proteases were hits, in addition to KLK8 (incl. ADAMTS10 that was discussed in rebuttal letter). As several other proteases are able to process activin A as well, is this a case of more general (promiscuous, serendipitous?) processing of pro-activin A by extracellular proteases rather than truly KLK8 specific one? Using a combination of genetic and biochemical approaches, we devoted altogether over 2x5 years of work to define the relative contributions of PCs versus PC-like protease activities, primarily to address this question of "specificity". Accordingly, we think that our findings already address this question in more depth than any related study in the TGF- β field:

- i. Our finding that in addition to Klk8, several proteases were required in parallel to enable PC-independent Activin-A *signaling* does not support the idea that they all cleave directly Activin-A, let alone serendipitously. In our view, a more likely scenario is that additional proteases act on additional regulators of this pathway, and/or in cascades. Concerning Adamts10, we were only asked whether a separate publication is expected to follow up on this candidate. Our answer that we are working on this does not imply that Adamts10

therefore does or should directly "...process activin A as well". Neither our manuscript nor our rebuttal letter claim that Adamts10 or any other candidate(s) from our RNAi screen *directly* cleave Activin-A. As already indicated also in the Discussion (p20), our working hypothesis is that this protease acts on additional candidate regulators of this pathway, such as fibrillins. To avoid a possible misunderstanding, our revised description of the top candidates from our RNAi screen on p6 now simply states: "Among the top candidates, especially *Klk8* significantly decreased the A70/A110 ratio in FurKO#1-βA cell SNs (Fig. 1G)." (i.e. without drawing special attention to ADAMTS10 or any other hits).

- ii. Among a panel of KLK8-related kallikreins, only the most closely related KLK5 mimicked KLK8 activity, thus further highlighting remarkable specificity.
- iii. Even if *some* overlap would exist with KLK5 or any other protease(s) at this level of *ex vivo* cleavage in *some* context, this would not contradict any of our claims. We also don't see why anyone should therefore question "biological/medical significance": KLK8 is not different in this regard from other proteases or kinases that are pursued as candidate drug targets. Finally, even if KLK8 were the only one (which we do not claim), we are not aware of a scientific method that could prove this comprehensively.
- iv. We clearly show *in vivo* and *in vitro* that both subunits of Activin-A *can* be cleaved independently of Klk8, at least by furin, and that even tumors depleted of both furin and Klk8 maintain circulating Activin-A, consistent with *in vitro* data that the precursor can also be cleaved cell non-autonomously. Importantly, however, both furin and cleavage by Klk8 were nevertheless essential specifically in signal-sending cells, and only in tumors expressing Activin-A. These observations highlight that Activin-A bioavailability and its oncogenic function are not simply determined by the net accumulation of cleaved forms by these or any other proteases, but by *how* it matures. For example, since most proteases have not only a single physiological substrate, an additional level of specificity should be expected to come from whether or not a given protease cleaves only Activin-A *alone*, or also some additional activin *regulatory* factor(s). At least for furin that is co-expressed with Activin-A *in cis*, we already know that it does indeed regulate also an Activin-A interacting factor.

When Nat. Commun. selected our manuscript for review and later for a revision, our initial findings already implied that future therapeutics targeting the stepwise cleavage must not ignore at least KLK5 as a potential additional player. To further highlight it, the revised Discussion now states on p13: "Moreover, incubation with KLK5 mimicked the ability of KLK8 to generate A70 in SNs containing A110, suggesting possible functional overlap in cells or tumors expressing both of these proteases, and/or that KLK5 might activate endogenous proKLK8." (revised text underlined). We believe that this should suffice to answer the remaining doubt of reviewer #2 given that neither this nor any other reviewer asked in their initial queries if processing could be "promiscuous". Reviewers #2-3 also do not request here to test this further. This is fair and means that it also should not be declared critical for our findings to reach "biological significance", especially after we already answered all questions, and after our new experiments already verified that the Activin-A processing intermediates are indeed no overexpression artifact, as requested by their actual queries.

Data from the negative control siRNA (siNT) shows on activin, again suggesting more unspecific effect at play here. We feel these doubts on biological significance (shared partly by the other reviewer) lower the impact and importance of this work.

While this particular control siRNA indeed diminished the baseline of A70 accumulation, it does not follow that this questions the specificity or the "biological significance" of KLK8:

1. siRNAi depletion was only one of several independent controls used to validate KLK8 specificity. For example, strong independent evidence comes from the fact that mutating R310 abrogates Activin-A activation by KLK8 but not by furin (Fig. 2F), and that this mimics the effect of Klk8 RNAi on activin-induced tumor growth (Fig. 6B).
2. Point 2 of our response to the initial query about siNT explained that while siNT happened to non-specifically reduce the number of cells, signal normalization to total

secreted Activin-A can control for such differences. Therefore, and since the number of cells was also noticeably diminished upon specific depletion of Klk8 (in this case by apoptosis, Fig. S4A) but not by just any siRNA, the non-negligible toxicity of this particular siNT sequence in our view should be regarded as an advantage rather than as a disadvantage, because it further reduces the risk of comparing apples to oranges.

3. In their initial comment, reviewers #2-3 stated that also siNT-treated control cells "...failed to process activin A correctly". Our data show that rather than being "incorrect", processing in siNT-treated cells was only less efficient. Even though siNT thus diminished the readout of our assay, our screen remained statistically very robust, as confirmed by its high Z' factor (Fig. 1E). Likewise, also in the subsequent validation experiments, the effect of Klk8 RNAi on A70 accumulation remained statistically significant *even when compared to this siNT*. Thus, our results obtained with siNT further corroborate the specificity of Klk8 rather than questioning it.
4. Nevertheless, to further document this, a new Fig. S1G of our revised manuscript now includes an additional independent experiment akin to the one in figure 1H, but where the gels were loaded with more protein to show that the clear difference between effects of siKlk8 vs siNT on A70 accumulation is highly reproducible. Moreover, for interested readers, the corresponding source data indicates that *Klk8* RNAi may also stabilize an Activin-A high molecular weight form. While this fits with our working model that KLK8 acts on proActivin-A in complex with an unknown interacting factor akin to LTBP (Pinjusic et al., in press), identifying this factor will be technically too challenging and too arduous for the scope of the present work. Accordingly, our revised manuscript does not elaborate on HMW forms, except for a newly added sentence in the Discussion where this model is now described and cited (p13, first paragraph): *"In a separate study, we recently found that cysteines C35 and C38 in the Activin-A prodomain form allosteric disulfide bonds with C314 and C322 that are essential to prevent precocious access of furin to the nearby penta-arginine motif before secretion, but that one β A subunit can bypass this inhibition, likely by interacting with a rate-limiting endogenous protein that remains to be identified³⁵."*
5. Lastly, RNAi toxicity was below detection in vivo and would not explain why two independent *Klk8* shRNAs specifically diminished only the growth of tumors that secrete Activin-A but not of control tumors that do not express this substrate. Finally, we showed that *Klk8* depletion selectively impaired cleavage of one subunit of Activin-A also in tumors, thus providing further independent confirmation that it inhibited tumor growth by *specifically* interfering with Activin-A processing.

...As we have not issues with technical aspects of the manuscript, we are happy to recommend this to be published in the revised form, but we are not sure if it quite at the level of significance to warrant Nature Communications.

We are very grateful that all four Nat. Commun. reviewers unanimously stated that this study should be published. Even reviewers #2-3 seem to only partly hesitate whether the "biological significance" warrants Nat. Commun. We do not see how this can mean anything else than that the actual claims of this study are substantiated (aside from a very sensible suggestion for only a very minor revision which we have now implemented in the revised manuscript, see below). Therefore, and since not even reviewers #2-3 judged our work to be *decidedly* below Nat. Commun. standards, we believe that Nature Communications should accept this manuscript to faithfully reflect this outcome. After all, it was Nat. Commun. that selected this work for review and then for a major revision based on claims that have remained unchanged and which are substantiated. Furthermore, if technical barriers would not have precluded to further test the impact of KLK8 on Activin-A function in a human model as well, we believe that the novelty of these findings would have warranted consideration by an even higher impact journal.

The only revision we would recommend authors to make is to do with the statement on apparent molecular weight of activins changing depending on acrylamide concentration in

SDS-PAGE (**page 18, lines 627-628**). Migration of proteins change on different percentage acrylamide, but proteins keep running in relation to their MW (or rather, hydrodynamic radius). There are protein-specific effect, e.g. with overall charge, but the apparent MW does not change from gel to gel. Otherwise MW markers would need to be different for different percentage gels. So, unless the authors have data showing convincingly that activin A changes its apparent MW (rather than migration) in different acrylamide concentrations or that the whole premise of SDS-PAGE since Laemmli is proven wrong, **we would recommend to re-phrase or even remove the statement: “apparent molecular weight that varies depending on the acrylamide concentration” (lines 627–628).**

We removed this likely overstatement and are grateful to the reviewer for pointing it out! Due to our anecdotal impression that the apparent MW of A30 varied on non-reducing gels with different concentrations of acrylamide, we probably too naively assumed that this is a general phenomenon, at least for such relatively small and presumably charged proteins. However, we never recorded or investigated this systematically enough to make this claim.

Reviewer #4 (Remarks to the Author):

I am satisfied with how the authors have responded to my concerns and have no further questions.